# ParaShield: Parameter-Level Directional Defense for Federated Backdoor Robustness

## Abstract

Heterogeneous federated learning improves the stealthiness of backdoor attacks, presenting substantial challenges for existing defense methods to simultaneously ensure effectiveness and robustness. However, divergent optimization objectives lead to pronounced parameter-level differences between the benign heterogeneous clients and those infected with backdoor attacks. To address this issue, we introduce Parameter-level Directional Defense, termed ParaShield, which leverages Neural Influence Factors (NIF) to dynamically and rapidly capture the critical parameters. ParaShield enables the identification of parameters that are essential for maintaining model performance within the benign client updates. On this basis, we further calculate the Cosine Similarity of Critical Parameters (CPCS) and the Sign Consistency of Critical Parameters (CPSC) to quantify directional alignment across client updates. Specifically, we initially filter out malicious model updates by analyzing the directional information of the critical parameters. Subsequently, we leverage the Mahalanobis distance in the 2D feature space formed by CPCS and CPSC to identify malicious updates deviating from the normal distribution, achieving robust aggregation. To comprehensively evaluate the robustness of ParaShield, we also construct the Projected Directional Backdoor Attack (PDBA), a stealthy backdoor attack that effectively examines defense mechanisms under realistic conditions. Extensive experiments conducted on various challenging Non-IID scenarios demonstrate the effectiveness of ParaShield.

## 1 Introduction

Federated learning McMahan et al. (2017); Kairouz et al. (2021); Bonawitz et al. (2017); Yang et al. (2019) is defined as a distributed machine learning paradigm that enables multiple clients to collaboratively train a shared global model while preserving the privacy of their local data. Due to its decentralized training paradigm, federated learning is inherently vulnerable to various attacks Li et al. (2020b); Kairouz et al. (2021). Among these, federated backdoor attacks are recognized as a specific form of data poisoning Bagdasaryan et al. (2020); Bhagoji et al. (2019); Ning et al. (2022); Xia et al. (2023); Fang & Chen (2023). In such attacks, a trigger is initially embedded into a dataset controlled by the attacker. The global model provided by the server is then trained on this poisoned dataset, and the resulting malicious model update is subsequently uploaded to the server for aggregation, enabling manipulation of the model's output.

In a realistic scenario, data heterogeneity Hsu et al. (2019); Li et al. (2020a); Ma et al. (2022) leads to differences in the data distribution among clients. Consequently, there are also differences in the update directions among benign clients. Backdoor attacks compel a model to simultaneously learn the main task and the backdoor task. Both benign and malicious updates deviate from the ideal model update direction, which makes it challenging to distinguish them at the client level. Therefore, we propose to analyze the differences in updates at the level of model parameters.

We argue that divergent fitting objectives produce distinct parameter distributions between benign heterogeneous and backdoor distributions Huang et al. (2024b). The presence of a backdoor trigger forces the model to adjust specific parameters to achieve the malicious output, leading to abnormal updates in these parameters. In contrast, benign heterogeneous updates also exhibit differences due to data heterogeneity, but these arise naturally and reflect the characteristics of the local dataset. Consequently, the parameter distributions of benign and backdoor updates are distinct.

We propose the Neural Influence Factor (NIF) to identify critical parameters that significantly impact model performance. As illustrated in Figure 1 (left), we empirically identify the critical parameters selected by NIF, ranked according to their importance. It can be seen that the critical parameters exhibit significantly higher importance in benign heterogeneous updates compared to backdoor updates. Further, Figure 1 (right) presents 10 clients, where the first three are malicious, highlighting that the NIF cosine similarity among these malicious clients is particularly high. This occurs because backdoor attacks modify specific backdoor-related parameters, resulting in abnormal updates. Although benign heterogeneous updates also exhibit patterns in NIF cosine similarity, these patterns are clearly distinct from those of backdoor updates. Such observations suggest that the parameter distributions of benign heterogeneous updates and malicious backdoor updates are fundamentally distinct.

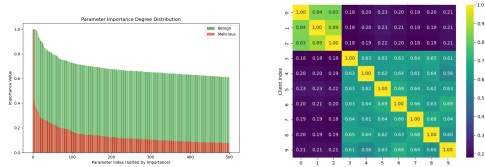

Figure 1: Motivation: Parameter importance degree distribution (left) shows the NIFs of critical parameters in benign and malicious model updates. Client NIF similarity (right) illustrates the NIF cosine similarity between different clients.

Inspired by AlignIns Xu et al. (2025), we leverage the Cosine Similarity of Critical Parameters (CPCS) and the Sign Consistency of Critical Parameters (CPSC) to quantify the differences between malicious and benign updates. By analyzing the update directions of critical parameters in benign heterogeneous updates, malicious model updates can be accurately identified. However, AlignIns struggles to maintain stable defense performance in the later stages when facing Badnet attacks in a non-IID setting. To mitigate unstable defense performance, we introduce the Adaptive Weighting Aggregation (AWA), which creatively combines CPCS and CPSC into a 2D feature distribution, and employs whitening transformation Ermolov et al. (2021) along with Mahalanobis distance Lee et al. (2018) to precisely adjust the weights of model updates, achieving robust aggregation. Figure 2 presents a comparison of defense performance between our method and AlignIns. It is evident that AlignIns suffers from unstable defense performance in later stages. Such instability results from the fixed threshold failing to effectively distinguish between benign and malicious updates as the model converges. In contrast, our method constructs a 2D feature distribution based on CPCS and CPSC, and applies a whitening transformation to remove potential correlations between features. By leveraging the Mahalanobis distance to identify malicious updates that deviate from the expected distribution, we adaptively adjust aggregation weights to achieve robust model aggregation.

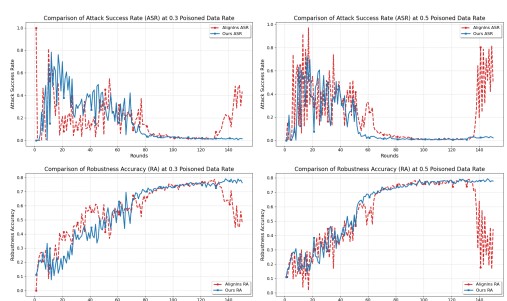

Figure 2: Comparison of Ours and AlignIns Performance

To rigorously assess ParaShield, we design a stealthy backdoor attack: Projected Directional Backdoor Attack (PDBA). We achieve dual stealth by controlling the gradient magnitude through projection and employing distributed triggers. Extensive experiments demonstrate that ParaShield maintains high accuracy while preserving robust defense capabilities in challenging Non-IID settings.

Our main contributions include:

- We propose ParaShield, a novel defense mechanism specifically designed to counteract backdoor attacks in heterogeneous federated learning. ParaShield imultaneously achieve high effectiveness and robustnes. Experimental results demonstrate the effectiveness of our method.

- We leverage NIF to rapidly and dynamically locate critical parameters in model updates. By analyzing the update directions of these critical parameters, we can precisely filter out malicious model updates.

- We design AWA, a robust aggregation strategy that constructs a high-quality feature distribution using critical parameters. AWA employs whitening transformation and Mahalanobis distance to adaptively allocate weights, effectively downweighting anomalous updates.

- We introduce Projected Directional Backdoor Attack (PDBA) as a stealthy backdoor attack. By constraining update magnitudes through gradient projection and employing distributed triggers, PDBA achieves dual stealth. Despite its evasive design, ParaShield successfully neutralizes PDBA, demonstrating the defense's superior detection capability under challenging threat models.

## 2 METHODOLOGY

We propose ParaShield, a novel backdoor defense framework tailored for heterogeneous federated learning, as illustrated in Figure 3 and detailed in Algorithm 1. The ParaShield framework consists of three primary modules: (1) Critical Parameter Extraction (CPE), dynamically identifies parameters that are most influential to model performance via Neural Influence Factor (NIF); (2) Critical Parameter-Based Filtering (CPF), filters malicious updates by leveraging Cosine Similarity of Critical Parameters (CPCS) and Sign Consistency of Critical Parameters (CPSC); (3) Adaptive Weighted Aggregation (AWA), assigns quality scores to client updates using whitening transformation and Mahalanobis distance to enable robust aggregation. To rigorously evaluate the robustness of ParaShield, we further design Projected Directional Backdoor Attack (PDBA), a novel attack characterized by dual stealth properties that simulate particularly challenging adversarial scenarios.

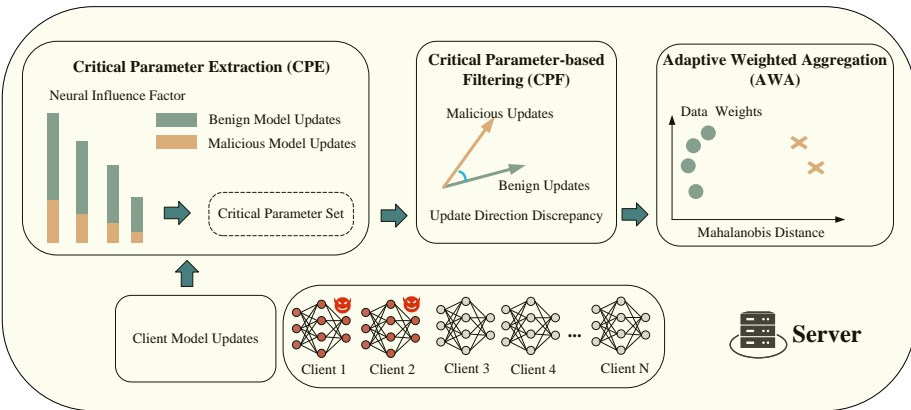

Figure 3: The Framework of ParaShield.

---

**Algorithm 1** ParaShield Aggregation

1: **Input**: Client model updates $\{\Delta w_{i,t}\}_{i=1}^n$, global model $w_{t-1}$, CPE ratio $\rho$, datasets $\{D_i\}$, AWA weight $\beta$
2: **Output**: Global model $w_t$
3: **for** $i \in [n]$ **do**
4:     Compute $NIF_i[j]$ by Eq. 1
5:     $I \leftarrow \text{RankTop}(NIF_i, \rho)$
6: **end for**
7: $S \leftarrow \emptyset$
8: **for** $i \in [n]$ **do**
9:     $CPSC_i \leftarrow \frac{1}{|I|} \sum_{j \in I} \left[ \text{sgn}(\Delta w_{i,t}[j]) == \text{sgn}\left(\sum_k \text{sgn}(\Delta w_{k,t}[j])\right)\right]$
10:     $CPCS_i \leftarrow \cos((\Delta w_{i,t} \odot I), w_{t-1})$
11:     Filter malicious model updates to update set $S$
12: **end for**
13: **for** $i \in S$ **do**
14:     $X_i \leftarrow [CPCS_i, CPSC_i], X_i' \leftarrow \text{whiten}(X_i)$
15:     $MD_i \leftarrow \sqrt{(X_i' - \mu)^T \Sigma^{-1} (X_i' - \mu)}, \mu = \text{mean}(\{X_i'\}), \Sigma = \text{cov}(\{X_i'\})$
16:     Convert $MD_i$ to quality score and compute weights using Eq. 6
17: **end for**
18: $w_t \leftarrow w_{t-1} + \sum_{i \in S} weight_i \cdot \Delta w_{i,t}$
19: **return** $w_t$

---

## 2.1 Critical Parameter Extraction

The parameter distributions of benign and backdoor updates in heterogeneous settings exhibit substantial differences. To capture these differences, we introduce Neural Influence Factor (NIF), which quantifies the contribution of individual parameters to model performance during federated updates. NIF adaptively assigns weights to parameters by combining their absolute update magnitude with their relative importance, thereby enhancing cross-client comparability. Formally, for client $i$ at round $t$, the NIF score for the $j$-th parameter is defined as:

$$NIF_{i,t}[j] = |\Delta w_{i,t}[j]| \times \frac{|\Delta w_{i,t}[j]| - \min(|\Delta w_{i,t}|)}{\max(|\Delta w_{i,t}|) - \min(|\Delta w_{i,t}|) + \epsilon}, \tag{1}$$

where $|\Delta w_{i,t}[j]|$ is the absolute value of the update to the $j$-th parameter by client $i$ at round $t$, $\min(|\Delta w_{i,t}|)$ and $\max(|\Delta w_{i,t}|)$ are the minimum and maximum absolute update values among all parameters of client $i$ at round $t$, and $\epsilon$ is a small constant to prevent division by zero.

Based on Eq. 1, we rank parameters based on their NIF scores, and the proportion of selected critical parameters is controlled by a hyperparameter $\rho$. By jointly considering absolute update magnitude and relative importance, NIF provides a comprehensive measure of the parameter significance. This selection filters out noisy parameters and emphasizes those that truly influence model performance, thereby alleviating the impact of data heterogeneity.

## 2.2 Critical Parameter-based Filtering

We introduce Critical Parameter-Based Filtering (CPF), which leverages the critical parameter index set $I$ obtained from the CPE module to detect malicious updates. CPF evaluates client updates using the Cosine Similarity of Critical Parameters (CPCS) and the Sign Consistency of Critical Parameters (CPSC). Specifically, CPCS measures the directional consistency of client updates on critical parameters in $I$:

$$CPCS_i = \cos((\Delta w_{i,t} \odot I), w_{t-1}) \tag{2}$$

where $(\Delta w_{i,t} \odot I)$ isolates parameters in the update $\Delta w_{i,t}$ based on $I$, and $w_{t-1}$ represents the global model parameters from the previous round. The CPCS evaluates the directional alignment between client updates and the global model update over the critical parameter set. By emphasizing directional consistency, CPCS filters out updates dominated by redundant parameters.

In federated learning, cosine similarity Huang et al. (2023); Fang & Chen (2023) is widely employed to assess the directional alignment of local model updates across clients. However, the contributions of individual parameters within a model update can differ substantially. Some redundant parameters exert minimal influence on overall model performance and may even introduce noise that impairs model accuracy. Therefore, CPCS is specifically designed to evaluate the directional consistency of critical parameters. An initial filtering of model updates is performed by comparing the update directions of clients' critical parameters with the overall direction of the global model update.

CPSC assesses the sign consistency of critical parameters in $I$:

$$CPSC_i = \frac{1}{|I|} \sum_{j \in I} [\text{sgn}(\Delta w_{i,t}[j]) == \text{sgn}(\sum_k \text{sgn}(\Delta w_{k,t}[j]))] \tag{3}$$

where $\text{sgn}(\Delta w_{i,t}[j])$ is the sign of the update to the $j$-th parameter by client $i$ at round $t$, and $\text{sgn}(\sum_k \text{sgn}(\Delta w_{k,t}[j]))$ denotes the majority sign direction across all clients $k$. Sign consistency reflects client consensus on the global model's optimization direction, helping detect malicious updates with directional deviations.

The Directional Manipulation Anomaly Score (DMAS) quantifies the anomaly level of client updates: $DMAS_i = \frac{|x_i - \text{Med}(X)|}{\sigma}$, where $x_i$ is the CPCS or CPSC score for client $i$, $\text{Med}(X)$ is the median score across all clients, and $\sigma$ is the standard deviation. Client updates with $DMAS_i$ exceeding a threshold $\tau$ are flagged as malicious and excluded from global model aggregation, enhancing federated learning robustness.

## 2.3 Adaptive Weighted Aggregation

To improve the robustness of model aggregation in federated learning, we propose the Adaptive Weighted Aggregation (AWA), which assesses the quality of each client's model update using

whitening transformation and Mahalanobis distance Huang et al. (2023). AWA leverages CPCS and CPSC to capture the directional and sign consistency of updates at the parameter level. Specifically, AWA can be further decomposed into several key steps.

First, we construct a feature vector for each client, defined as $X_i = [CPCS_i, CPSC_i]$, to represent the quality of its model update. CPCS measures the directional consistency of client updates on the critical parameters index set, while CPSC evaluates the consistency of critical parameter signs with the dominant trend. These metrics form a high-quality feature distribution for subsequent analysis.

Next, we apply a whitening transformation to eliminate correlations and scale differences among feature dimensions caused by data heterogeneity. In high-dimensional vector spaces, correlations among feature dimensions and differences in scale can render direct distance calculations ineffective. We address this issue by applying whitening transformation of client feature vectors. The whitening process decorrelates the features, resulting in more independent dimensions and thereby enhancing the effectiveness and discriminative capability of the Mahalanobis distance. The whitening transformation is defined as:

$$X_i' = (X_i - \mu) \cdot \Sigma^{-1/2} \tag{4}$$

Here, $\mu = \text{mean}(\{X_i\})$ denotes the mean vector of all client feature vectors. Additionally, $\Sigma = \text{cov}(\{X_i\})$ represents the covariance matrix of the feature vector set. The inverse square root of the covariance matrix, $\Sigma^{-1/2}$, is computed via eigenvalue decomposition. This transformation produces a whitened feature vector $X_i'$, ensuring that features are decorrelated and have unit variance.

By incorporating statistical covariance information, the Mahalanobis distance effectively identifies malicious updates that retain anomalous characteristics even after whitening. Subsequently, we compute the Mahalanobis distance to assess the quality of each client's update:

$$MD_i = \sqrt{(X_i' - \mu)^T \Sigma^{-1} (X_i' - \mu)} \tag{5}$$

The Mahalanobis distance incorporates the covariance matrix $\Sigma^{-1}$ to account for the joint distribution of features, measuring the deviation of a client's update from the group norm in the original feature space. A larger $MD_i$ indicates a significant deviation, potentially signaling a malicious update. The whitening transformation enhances the discriminative power of the Mahalanobis distance by eliminating feature correlations and scale disparities, which can render direct distance metrics ineffective in high-dimensional spaces.

The unbounded Mahalanobis distance is then mapped to a normalized trust score $Q_i$, using a negative exponential function. Finally, the aggregation weight for each client is computed by combining the trust score with the client's data contribution:

$$weight_i = \beta \cdot Q_i + (1 - \beta) \cdot D_i \tag{6}$$

where $Q_i$ denotes the quality-based trust score, $D_i$ represents the client's proportion of the total data, and $\beta \in [0, 1]$ is a hyperparameter balancing quality and quantity. By performing weighted aggregation, AWA prioritizes high-quality updates while accounting for data contributions, achieving robust defense against malicious updates.

## 2.4 PROJECTED DIRECTIONAL BACKDOOR ATTACK

To evaluate ParaShield, we propose the Projected Directional Backdoor Attack (PDBA), a stealthy attack that integrates distributed backdoor triggers with Projected Gradient Descent (PGD) to obscure both update magnitude and direction. PDBA decomposes the backdoor trigger into disjoint components, each injected by malicious clients into their local data during training. This results in updates without detectable magnitude anomalies, allowing the backdoor to gradually manifest in the global model through aggregation, thereby evading conventional detection mechanisms.

PGD constrains updates within an $\ell_2$-norm ball of radius $\delta$, achieving magnitude stealth by matching the update norm to benign clients and directional stealth by aligning update directions with benign updates. These constraints enable the attack to evade both norm-based and outlier detection. The projected parameter vector is:

$$w_i^{t,\text{proj}} = \begin{cases} w_i^t, & \text{if } \|w_i^t - w_{t-1}\|_2 \leq \delta \\ w_{t-1} + \delta \cdot \frac{w_i^t - w_{t-1}}{\|w_i^t - w_{t-1}\|_2}, & \text{otherwise} \end{cases} \tag{7}$$

where $w_i^t$ is the local model parameter after training with poisoned data, $w_{t-1}$ is the global model parameter at round $t$, and $\delta$ is the $\ell_2$ norm threshold.

By combining distributed triggers with PGD, PDBA enables stealthy backdoor accumulation in the global model while preserving main-task accuracy, posing a strong challenge to federated learning defenses.

## 3 EXPERIMENT SETUP

**Datasets and Models:** The experimental evaluation is conducted on two widely used benchmark datasets: CIFAR-10 Krizhevsky et al. (2009) and CIFAR-100 Krizhevsky et al. (2009). For CIFAR-10, a ResNet-9 model is employed, whereas a ResNet-18 model He et al. (2016) is adopted for CIFAR-100.

**Baseline Defenses:** To ensure a comprehensive evaluation, the proposed method is benchmarked against eight representative baseline approaches, encompassing both robust aggregation strategies and backdoor defense mechanisms, including FedAvg McMahan et al. (2017), RFA Pillutla et al. (2022), RLR Ozdayi et al. (2021), Foolsgold Fung et al. (2020), MultiMetric Huang et al. (2023), Mkrum Blanchard et al. (2017), Trim Yin et al. (2018), and AlignIns Xu et al. (2025).

**Backdoor Attacks:** Four representative backdoor attack methods, including PDBA, Neurotoxin Zhang et al. (2022), DBA Xie et al. (2019), and Badnet Gu et al. (2017), are selected for evaluation.

**Evaluation:** The proposed defense mechanism is evaluated using three key metrics. **Main-task Accuracy (MA)** measures classification accuracy on the clean test dataset. **Attack Success Rate (ASR)** assesses the accuracy of classifying poisoned test samples with the trigger as the target class. **Robust Accuracy (RA)** evaluates classification accuracy on test samples with the backdoor trigger but original labels.

## 4 EXPERIMENT RESULTS

### 4.1 ROBUSTNESS AGAINST DIFFERENT ATTACKS AND COMPARISON WITH SOTA

The experiments were conducted in a simulated federated learning system comprising N=20 clients. Unless otherwise specified, the Poisoned Model Rate (PMR) was set to 0.2, corresponding to 4 of the 20 clients being malicious. For these malicious clients, the Poisoned Data Rate (PDR) was established at 0.5, signifying that 50% of their local data was poisoned for the backdoor task. Data heterogeneity was modeled by partitioning the dataset among all clients using a Dirichlet distribution, where the degree of Non-IID data was set to $\alpha = 0.5$. The proposed method, ParaShield, was compared with the SOTA method through extensive experiments on the CIFAR-10 and CIFAR-100 datasets. The comprehensive results are presented in Table 1, where the best-performing results are highlighted in bold and the second-best are underlined.

On the CIFAR-10 dataset, only RLR Ozdayi et al. (2021), AlignIns Xu et al. (2025), and the proposed ParaShield method achieved low ASR when evaluated against four backdoor attacks. This finding suggests that methods relying solely on cosine similarity, such as MultiMetric Huang et al. (2023) and Foolsgold Fung et al. (2020), fail to effectively detect these sophisticated attacks. Other baseline defenses were also observed to exhibit significant limitations. For instance, while an ASR of only 4.20% was recorded for Mkrum Blanchard et al. (2017) against PDBA, the rate increased dramatically to 94.44% under Neurotoxin Zhang et al. (2022), revealing its vulnerability to parameter-specific directional manipulations. Although RLR Ozdayi et al. (2021) achieved a low ASR, its RA remained significantly inferior to that of the proposed ParaShield. Furthermore, general-purpose robust aggregation methods, such as RFA Pillutla et al. (2022) and Trim Yin et al. (2018), were also found to be ineffective.

The primary comparison was conducted with AlignIns. Against the PDBA attack, ASR was further reduced by 2.95%, and RA was increased by 4.75% by ParaShield in comparison to AlignIns. Additionally, when facing the parameter-manipulating Neurotoxin Zhang et al. (2022) attack, a targeted

Table 1: Performance comparison of different defense methods against various backdoor attacks on CIFAR-10 and CIFAR-100 datasets.

| Dataset | Defense | PDBA | | | Neurotoxin | | | DBA | | | Badnet | | |
|---|---|---|---|---|---|---|---|---|---|---|---|---|---|
| | | ASR ↓ | RA ↑ | MA ↑ | ASR ↓ | RA ↑ | MA ↑ | ASR ↓ | RA ↑ | MA ↑ | ASR ↓ | RA ↑ | MA ↑ |
| CIFAR-10 (ResNet 9) | FedAvg | 82.83 | 15.87 | 86.62 | 88.70 | 10.80 | 85.08 | 73.00 | 24.68 | 86.37 | 90.57 | 9.21 | 86.42 |
| | RLR | 2.09 | 55.71 | 57.04 | **1.33** | 53.89 | 53.97 | 1.10 | 61.10 | 62.29 | 3.04 | 55.33 | 57.84 |
| | RFA | 93.60 | 5.60 | 81.39 | 70.86 | 24.63 | 79.87 | 84.37 | 15.36 | 79.56 | 94.74 | 4.67 | 79.95 |
| | Foolsgold | 60.37 | 34.88 | **86.37** | 2.80 | 80.78 | 84.26 | 71.20 | 26.33 | **86.18** | 84.53 | 14.66 | **85.97** |
| | MultiMetric | 69.44 | 26.37 | 82.13 | 98.83 | 1.06 | 81.71 | 75.07 | 22.72 | 82.59 | 96.89 | 2.91 | 82.32 |
| | Mkrum | 4.20 | 78.33 | 81.46 | 94.44 | 4.80 | 78.34 | 4.07 | 78.62 | 81.07 | 97.42 | 2.37 | 78.76 |
| | Trim | 86.97 | 11.51 | 85.54 | 48.98 | 44.57 | 84.03 | 88.56 | 10.40 | 85.48 | 84.56 | 14.19 | 85.20 |
| | AlignIns | 3.78 | 78.09 | 82.99 | 1.91 | 79.39 | 81.91 | 2.68 | 81.41 | 83.78 | 1.73 | 81.56 | 83.89 |
| | Ours | **0.83** | **82.84** | 84.56 | 1.87 | **82.01** | **84.92** | **0.83** | **82.22** | 84.67 | **1.14** | **82.24** | 84.30 |
| CIFAR-100 (ResNet 18) | FedAvg | 99.87 | 0.13 | 67.17 | 95.25 | 4.01 | 66.38 | 99.74 | 0.25 | 66.63 | 99.90 | 0.10 | 66.89 |
| | RLR | 34.19 | 21.67 | 36.43 | 0.55 | 17.54 | 18.11 | 18.86 | 27.87 | 35.89 | 99.40 | 0.41 | 35.95 |
| | RFA | **0.22** | 38.34 | 41.05 | 3.48 | 35.21 | 41.97 | 0.47 | 36.90 | 39.52 | 0.55 | 38.66 | 42.11 |
| | Foolsgold | 99.75 | 0.24 | 66.27 | 0.47 | 61.68 | 65.50 | 99.76 | 0.23 | 66.30 | 99.77 | 0.23 | 66.11 |
| | MultiMetric | 100.00 | 0.00 | 46.75 | 99.95 | 0.05 | 41.21 | 99.96 | 0.04 | 46.05 | 99.92 | 0.07 | 45.50 |
| | Mkrum | 98.36 | 1.41 | 60.97 | 96.04 | 0.17 | 60.98 | 97.75 | 1.79 | 60.68 | 98.38 | 1.46 | 61.28 |
| | Trim | 99.78 | 0.22 | **66.91** | 87.70 | 9.28 | 65.22 | 99.88 | 0.11 | **67.01** | 99.94 | 0.06 | 66.41 |
| | AlignIns | 0.34 | 59.23 | 63.65 | 0.71 | 57.18 | 60.52 | 0.51 | 56.12 | 59.63 | 0.67 | 56.84 | 60.39 |
| | Ours | 0.30 | **60.65** | 65.17 | **0.31** | **61.92** | **66.45** | **0.20** | **60.20** | 64.73 | **0.36** | **61.95** | **66.67** |

direction check on critical parameters enabled ParaShield to achieve the highest observed RA of 82.01%.

On the more complex CIFAR-100 dataset, where increased task difficulty leads to a general reduction in MA across all methods, an even more pronounced and consistent advantage was demonstrated by ParaShield over AlignIns in both robustness and overall performance. Specifically, improvements in RA and MA were observed with ParaShield: 1.42% and 1.52% against the PDBA attack, 4.74% and 5.93% against the Neurotoxin attack, 4.08% and 5.11% against the DBA Xie et al. (2019) attack, and 5.11% and 5.98% against the Badnet Gu et al. (2017) attack. These results collectively indicate that superior protection is provided by ParaShield across a wide range of backdoor attacks, while better preservation of model utility is also achieved, particularly on complex tasks.

## 4.2 VISUAL ANALYSIS OF THE AWA MODULE

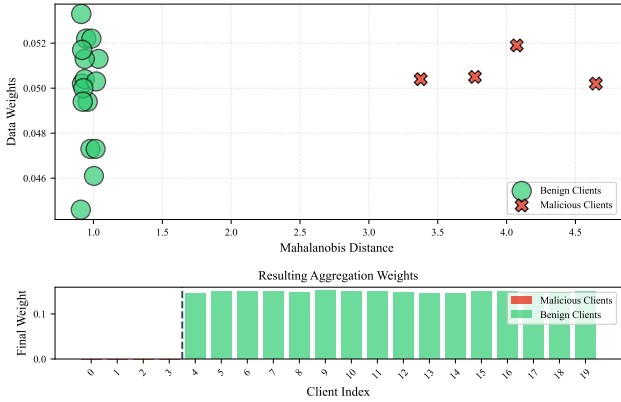

Figure 4: Visual Analysis of the AWA Module based on Mahalanobis Distance

We visualized the defensive effect of AWA using the training results from a specific round in ParaShield. The default experimental configuration includes 20 clients, of which 4 are malicious. We constructed a high-quality feature distribution from the cosine similarity and sign consistency of critical parameters, followed by a whitening transformation to decorrelate the features. The most critical component of this process is the measurement of the feature distribution's dispersion via Mahalanobis distance. Figure 4 shows that the feature distribution of the four malicious clients is

distinctly separate from the distribution of the benign clients. In the event that CPF does not successfully screen malicious client updates, AWA ensures robust aggregation by assigning negligible weights to them. As shown in Figure 4, CPF has already successfully filtered out these malicious model updates. Consequently, in the aggregation phase, ParaShield successfully discarded all malicious model updates. This ensures that weights are assigned exclusively to benign updates, thereby achieving robust aggregation.

### 4.3 IMPACT OF TRAINING ENVIRONMENT

#### 4.3.1 IMPACT OF THE POISONED DATA RATE

Poisoned Data Ratio (PDR) represents the proportion of poisoned samples relative to the total sample size. In this experiment, the PDR was varied across a wide range from 0.1 to 0.9. We evaluated ParaShield and various defenses on the CIFAR-100 dataset under various attacks, using ASR and RA as evaluation metrics.

As shown in Figure 5, ParaShield's defensive performance is consistently strong and stable across all tested PDR levels, highlighting its robustness against varying attack intensities.

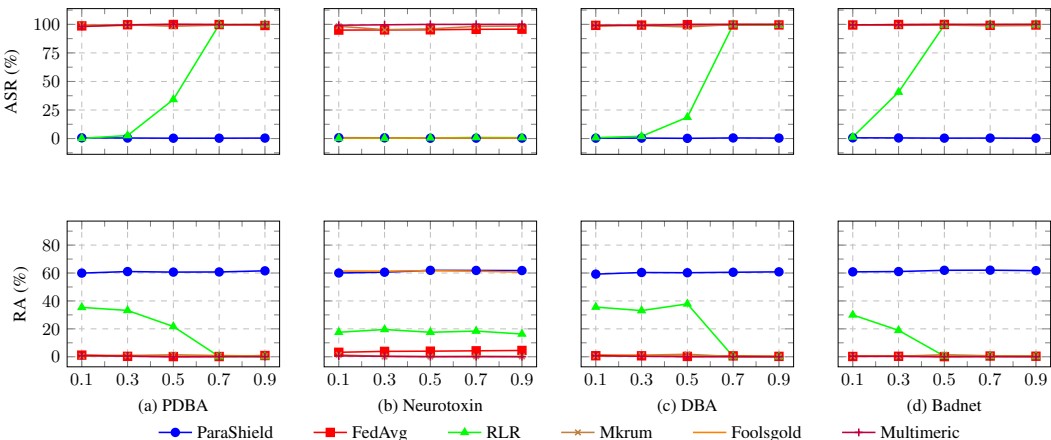

Figure 5: Comparison of ParaShield and various defenses under varying PDR values. Top row: ASR; bottom row: RA.

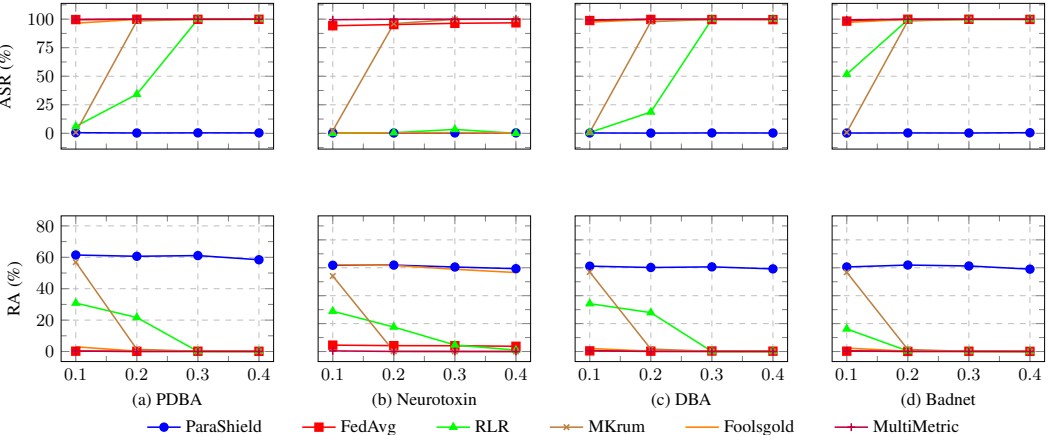

Figure 6: Comparison of ParaShield and various defenses under varying PMR values. Top row: ASR; bottom row: RA.

### 4.3.2 Impact of the Poisoned Model Rate

Poisoned Model Ratio (PMR) represents the proportion of malicious clients among all participating clients. In a simulated environment consisting of 20 clients, the number of attackers was varied from 2 to 8, corresponding to a PMR range of 0.1 to 0.4. We further compared ParaShield with various defenses on CIFAR-100 under various attacks, using ASR and RA as evaluation metrics.

Figure 6 demonstrates that ParaShield maintains excellent defensive capabilities across all tested PMR values. Notably, ParaShield remains highly effective against the PDBA attack even at a PMR as high as 0.4.

## 4.4 Ablation Study

We conducted a comprehensive ablation study to validate the individual contributions of CPE, CPF, and AWA modules in defending against federated backdoor attacks. The study was conducted on the CIFAR-100 dataset under a Badnet attack, sequentially removing each of the three modules while maintaining a PDR of 0.3. The results are summarized in Table 2.

The analysis demonstrates that CPF, as a parameter-level method for inspecting model update directions, constitutes the most critical component of ParaShield. CPE focuses on critical parameters in model updates to effectively mitigate interference caused by data heterogeneity. AWA serves as an essential complement to CPF by adaptively adjusting the weights of model updates.

Table 2: Ablation study of the ParaShield components on the CIFAR-100 dataset under the Badnet attack.

| Module Configuration | ASR (%) ↓ | RA (%) ↑ |
|---|---|---|
| ParaShield (without CPE) | 32.56 | 57.70 |
| ParaShield (without CPF) | 54.13 | 41.67 |
| ParaShield (without AWA) | 50.43 | 44.26 |
| ParaShield (Full Model) | **1.64** | **77.59** |

## 5 Related Works

Backdoor attacks threaten federated learning and have prompted the latest defenses in robust aggregation, client-side detection .

**Robust Aggregation and Outlier Detection.** Robust aggregation mitigates malicious client impact. Fisher Calibration Huang et al. (2024a) and Parameter Disparities Dissection Huang et al. (2024b) use the Fisher Information Matrix to compare local and global parameter importance, downweighting suspicious updates with large discrepancies. Scope Huang et al. (2025) focuses on update direction. Scope targets metric-constrained attacks by amplifying backdoor gradient dimensions through normalization and differential scaling.

**Client-Side and Proactive Defenses.** Client-side methods empower clients in defense. BackdoorIndicator Li & Dai (2024) uses Out-of-Distribution data to compute parameter importance, identifying malicious clients via clustering of outlier importance vectors. Snowball Qin et al. (2024) employs a collaborative election-based approach, where clients vote on peer updates to filter malicious ones from a decentralized perspective. More details are in Appendix.

## 6 Conclusion

This paper proposes ParaShield, a novel backdoor defense method in heterogeneous federated learning. ParaShield dynamically tracks critical parameters in model updates and conducts parameter-level directional anomaly detection. We also introduce the utilization of parameter-level directional information to construct high-quality features for adaptive weighting. To comprehensively assess ParaShield's resilience, we propose PDBA as a stealthy backdoor attack. Experimental results conducted under challenging Non-IID conditions demonstrate the effectiveness of our method.

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

# A APPENDIX

## A.1 THE USE OF LARGE LANGUAGE MODELS

The language of this paper was polished using large language models (LLMs) to enhance clarity and readability. The final content and academic integrity remain the responsibility of the authors.

## A.2 IMPACT OF DIFFERENT DEGREES OF NON-IID

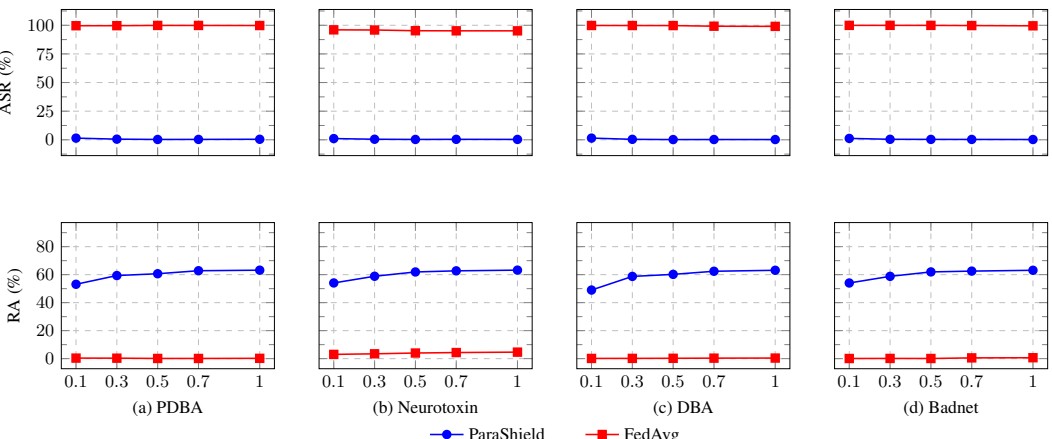

Figure 7: Comparison of ParaShield with Fedavg under different degrees of Non-IID. Top row: ASR; bottom row: RA.

To validate the robustness of the proposed ParaShield framework under varying degrees of data heterogeneity, an experiment was conducted on the CIFAR-100 dataset. The performance of ParaShield was compared with that of the FedAvg baseline McMahan et al. (2017) under various attacks, with both ASR and RA evaluated. The degree of Non-IID was controlled by the Dirichlet parameter, $\alpha$, which was varied from 0.1 to 1.0.

As illustrated in Figure 7, a consistently low ASR and a high, stable RA are maintained by ParaShield across the entire range of $\alpha$ values. These results highlight ParaShield's excellent generalization capabilities and consistent efficacy across diverse data distributions.

## A.3 IMPACT OF HYPERPARAMETERS

### A.3.1 IMPACT OF THE CRITICAL PARAMETER EXTRACTION RATIO

The selection ratio of critical parameters in the CPE is recognized to play a decisive role in the effectiveness of the subsequent CPF. A selection of too few critical parameters may result in a biased and one-sided judgment of the model update's direction. Conversely, an excessive selection undermines the fundamental purpose of CPE, as not all parameters contribute equally, and redundant parameters may negatively impact model performance.

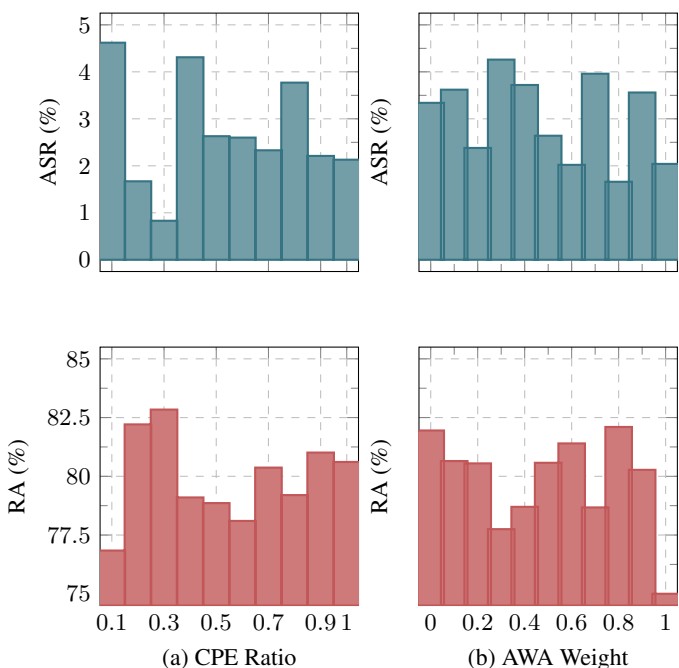

Figure 8: Sensitivity analysis of ParaShield under varying hyper-parameters. Left column: Top - ASR vs. Proportion, Bottom - RA vs. Proportion; Right column: Top - ASR vs. Weight, Bottom - RA vs. Weight.

Therefore, a study was conducted on the CIFAR-100 dataset to investigate the optimal selection ratio for critical parameters when defending against the PDBA attack. As illustrated in Figure 8 (a), depicting the impact of this ratio on the model's defensive performance, the optimal defensive performance is achieved at a selection ratio of 0.3.

### A.3.2 THE QUALITY-QUANTITY TRADE-OFF

The weighting between the quality score and data size in AWA is recognized as critical for determining each client's contribution to the global model. If aggregation is based solely on data size, as in the standard FedAvg McMahan et al. (2017), malicious updates may be assigned disproportionately large weights due to their volume, thus enabling manipulation of the model update direction. Conversely, exclusive reliance on the quality score while disregarding data size may lead to diminished generalization capabilities of the model. Therefore, identifying an optimal balance between these two factors is essential to maximize overall model performance.

To investigate the impact of this weighting, a study was conducted on the CIFAR-100 dataset under the PDBA attack. Figure 8 (b) illustrates the effect of varying this weighting factor on the model's defensive performance. The results indicate that optimal defense performance is achieved when the weight is set to 0.8. This finding suggests that a quality-dominant aggregation strategy produces the best outcomes.

### A.4 RELATED WORK

### A.4.1 FEDERATED LEARNING

Federated learning Yang et al. (2019) is defined as a distributed machine learning paradigm developed to address critical issues such as data privacy and data silo problems Kairouz et al. (2021). In this framework, a cohort of $n$ clients collaboratively train a shared global model $w$ using their respective local datasets. After each round of local training, model updates computed by the clients are transmitted to the central server, where aggregation is performed to refine the global model $w$. In

the $t$-th communication round of FedAvg McMahan et al. (2017), a subset of $k$ clients is randomly selected by the server to participate in the aggregation process.

$$w_t = w_{t-1} + \eta \cdot \frac{\sum_{i=1}^{k} D_i \Delta w_{i,t}}{\sum_{i=1}^{k} D_i} \tag{8}$$

where $w_t$ denotes the global model at the $t$-th round, $\Delta w_{i,t}$ refers to the local model update computed by the $i$-th client using its local dataset $D_i$ at the $t$-th round, and $\eta$ represents the learning rate.

### A.4.2 BACKDOOR ATTACKS IN FEDERATED LEARNING

Federated backdoor attacks Bagdasaryan et al. (2020); Bhagoji et al. (2019); Xie et al. (2019); Zhang et al. (2022); Ning et al. (2022); Xia et al. (2023); Fang & Chen (2023) are categorized into two primary types based on the characteristics of model updates: magnitude-based and direction-based attacks. Magnitude-based attacks are characterized by model updates from malicious clients that exhibit significantly larger or smaller numerical magnitudes compared to those from benign clients. Specific parameter updates are amplified by attackers to dominate global model aggregation and embed a backdoor. A notable example is the Model Replacement Attack Bagdasaryan et al. (2020), in which local model updates are scaled by malicious clients by a large factor to overwrite the global model with a malicious version, effectively implanting the backdoor.

Direction-based attacks, in contrast, maintain updates of normal magnitude, making them difficult to detect through numerical analysis alone. The update direction is manipulated to deviate significantly from the average direction of benign clients, thereby steering the global model toward the attacker's backdoor objective. This approach poses particular challenges in Non-IID settings Hsu et al. (2019), where natural directional variations among client updates further complicate detection. The Distributed Backdoor Attack (DBA) Xie et al. (2019) utilizes distributed triggers across multiple malicious clients to coordinate and control the update direction of the global model. In F3BA Fang & Chen (2023), the sign of specific parameters is inverted, causing the local update to diverge from the majority while maintaining a normal magnitude.

A subcategory of direction-based attacks, known as masked or sparsity-based attacks, enhances stealth by injecting backdoor updates into a small subset of key parameters or features, while leaving most parameters nearly unchanged. Neurotoxin Zhang et al. (2022) targets stable model parameters characterized by low variance during training, embedding the backdoor within these regions to ensure persistence. This approach increases the durability of the attack and reduces its susceptibility to detection and mitigation during global aggregation.

### A.4.3 BACKDOOR DEFENSES IN FEDERATED LEARNING

Backdoor defenses in Federated Learning Blanchard et al. (2017); Yin et al. (2018); Cao et al. (2020); Ozdayi et al. (2021); Nguyen et al. (2022); Huang et al. (2023) can be broadly classified into three primary categories.The first category is aimed at mitigating the impact of backdoors. Existing methods in this category typically involve the addition of statistical noise Geyer et al. (2017) to the updates or the application of norm clipping Sun et al. (2019). While straightforward, these approaches inevitably result in performance degradation of the global model, leading to reduced accuracy.

The second category is characterized by statistics-based robust aggregation, in which statistical metrics are employed to combine model updates in a more secure manner. For example, in RFA Pillutla et al. (2022), aggregation is performed using the geometric median of all model updates, while in Trim Yin et al. (2018), a certain fraction of the most extreme parameter values is discarded prior to calculating their arithmetic mean for the final update.

The third category is composed of filtering-based detection methods Shejwalkar & Houmansadr (2021); Awan et al. (2021), in which malicious contributions are identified and excluded prior to aggregation.These methods can be further categorized into distance-based and direction-based approaches. For example, in Mkrum Blanchard et al. (2017), the squared Euclidean distance among all client updates is calculated in order to select a subset for aggregation.In direction-based approaches, cosine similarity is often utilized. For instance, in Foolsgold Fung et al. (2020), a similarity matrix is

computed among updates to adjust the aggregation weight assigned to each client. Hybrid methods, such as MultiMetric Huang et al. (2023), employ a combination of Euclidean distance, L2 norm, and cosine similarity to identify and filter out malicious clients.It is important to note, however, that directional backdoor attacks are capable of easily bypassing basic distance- and direction-based detection methods.

