# OpenReview forum: "ParaShield: Parameter-Level Directional Defense for Federated Backdoor Robustness"
_ICLR.cc/2026/Conference — Submitted to ICLR 2026_

### Official Review · Reviewer_Vyao · 2025-10-17

**Soundness:** 2
**Presentation:** 3
**Contribution:** 2
**Rating:** 4
**Confidence:** 4

**Summary:**

This paper proposes ParaShield, a parameter-level directional defense method for heterogeneous federated learning. It identifies critical parameters through a Neural Influence Factor (NIF), filters malicious updates using cosine and sign consistency measures (CPCS/CPSC), and employs Adaptive Weighted Aggregation (AWA) with whitening and Mahalanobis distance for robust model aggregation. The authors also propose a stealthy Projected Directional Backdoor Attack (PDBA) to evaluate defense strength. Experiments on CIFAR-10/100 show ParaShield achieves the lowest ASR and highest robustness among tested defenses.

**Strengths:**

- **Well-motivated framework**. The paper addresses heterogeneity-induced difficulties in distinguishing benign and backdoored updates.
- **Comprehensive evaluation**. The paper includes multiple backdoor types, Non-IID settings, and ablation studies.
- **New attack**. The paper adds value for testing robustness under realistic threat models.

**Weaknesses:**

- No comparison with clustering- or influence-based defenses such as DeepSight [1] or FLAME [2], which are relevant for aggregation-level defenses.
- Limited methodological novelty: The main difference from AlignIns is the adaptive weighting (AWA). The large performance gain lacks clear interpretability or theoretical justification.
- Lack of interpretability: The mechanism behind the improvement from CPCS/CPSC + whitening is not analyzed or visualized.
- No runtime/memory analysis: The computational cost of each module (CPE, CPF, AWA) is not reported.
- Scalability not demonstrated: Only small CNN backbones and simple FL setups are tested. No evidence on larger models (e.g., ViT) or real-world-scale scenarios.
- Source code not released, limiting reproducibility and community validation.

[1] Rieger, Phillip, et al. "DeepSight: Mitigating Backdoor Attacks in Federated Learning Through Deep Model Inspection."\
[2] Nguyen, Thien Duc, et al. "{FLAME}: Taming backdoors in federated learning." 31st USENIX Security Symposium (USENIX Security 22). 2022.

**Questions:**

- Could you provide comparisons with clustering- or influence-based defenses such as DeepSight or FLAME, to better position ParaShield among aggregation-level defense methods?
- Could you explain why the adaptive weighting (AWA) module leads to such a large performance gain over AlignIns, given the architectural similarity between the two frameworks?
- Could you provide runtime and memory analyses for each module (CPE, CPF, AWA) to better understand the computational overhead of ParaShield?
- Do you plan to release the source code, or could you share implementation details to ensure reproducibility and independent validation of your results?

---

> ### Author Response · Authors · 2025-11-21
> **Thanks for the Reviewer Vyao's valuable comments, we will address all the questions as follow.**
>
> > #### [Q1]: Could you provide comparisons with clustering- or influence-based defenses such as DeepSight or FLAME, to better position ParaShield among aggregation-level defense methods?
>
> > #### Thank you for raising this question. In our experimental scenario, Flame exhibits an average runtime per round of 4.6578s, whereas our proposed method averages only 0.2204s per round. We tested Flame against the PDBA attack using the CIFAR-10 dataset. Flame resulted in an ASR of 0.5860 and an RA of 0.3264. In contrast, our method achieved an ASR of 0.0083 and an RA of 0.8284. Clearly, Flame is significantly inferior to our method in terms of both computational overhead and defense performance.
>
> > #### [Q2]: Could you explain why the adaptive weighting (AWA) module leads to such a large performance gain over AlignIns, given the architectural similarity between the two frameworks?
>
> > #### Thank you for raising this question. Leveraging CPE (Critical Parameter Extraction) to rapidly identify critical model parameters, we then employ CPF to analyze their directional information. The introduction of AWA successfully addresses the unstable defense efficacy associated with threshold-dependent methods, such as AlignIns, ultimately resulting in low ASR and high RA.
>
> > #### [Q3]: Could you provide runtime and memory analyses for each module (CPE, CPF, AWA) to better understand the computational overhead of ParaShield?
>
> > #### Thank you for raising this question. In our experimental setup, we measured the average runtime per communication round for ParaShield and several state-of-the-art baselines under identical hardware and software environments. The results are summarized below.
> >
> > #### ParaShield introduces two additional lightweight components: CPE (based on simple parameter-wise ranking) and  CPF/AWA . These steps incur only linear complexity with respect to the model size, and the Mahalanobis distance is computed in a 2-dimensional space, making its cost negligible.
> >
> > #### As shown in our runtime measurements, ParaShield exhibits comparable per-round computational overhead to existing  detection-based approaches , and is significantly more efficient than aggregation methods that rely on high-dimensional distance computations or iterative geometric median calculations.  We will add the detailed runtime table and clarify this comparison in the revised manuscript.
> >
> > | Method      | Avg. Runtime (s) |
> > | ----------- | ---------------- |
> > | Flame       | 4.6578           |
> > | Foolsgold   | 1.1667           |
> > | RFA         | 15.6842          |
> > | Fedavg      | 0.0013           |
> > | Trim        | 0.0372           |
> > | AlignIns    | 0.1261           |
> > | Mkrum       | 0.1869           |
> > | MultiMetric | 9.7010           |
> > | Ours        | 0.2204           |
>
> > #### [Q4]: Do you plan to release the source code, or could you share implementation details to ensure reproducibility and independent validation of your results?
>
> > #### Thank you for raising this question. We have included the algorithm's pseudocode for our method in the supplementary material and we plan to upload the full source code at a later date.

---

> > ### Comment · Reviewer_Vyao · 2025-11-27
> > **Acknowledge to read the author's response**
> >
> > Thank you for the additional clarifications and runtime comparisons. However, two key concerns remain insufficiently addressed.
> > First, the distinction between ParaShield and AlignIns is still unclear: although AWA is presented as the main improvement, the overall pipeline (critical parameter identification + directional filtering + weighting) appears highly similar, and it is not yet explained why AWA alone leads to such a large performance gain. Second, the mechanism behind CPCS/CPSC combined with whitening is still largely heuristic, and the paper would benefit from clearer intuition or visualization showing how these steps separate benign and malicious updates under heterogeneity. As these issues relate to the core methodological novelty and interpretability, my evaluation remains slightly below the acceptance threshold, and I would appreciate further clarification from the authors.

---

> ### Author Response · Authors · 2025-11-27
> **Thank you for your response. We have supplemented the Theoretical Analysis: κ-Robustness of ParaShield.**
>
> # Theoretical Analysis: $\kappa$-Robustness of ParaShield
>
> We theoretically analyze the robustness of ParaShield, a composite federated backdoor defense that first performs Critical Parameter Extraction (CPE), followed by high-precision filtering via Critical Parameter-Based Filtering (CPF) and Adaptive Weighted Aggregation (AWA).
>
> ## Standard Assumptions on Benign Clients
>
> We adopt the following standard assumptions on the local updates from benign clients $\mathcal{B}$ (as used in Krum, Trimmed Mean, FLTrust, etc.):
>
> - Assumption 1 ($\mu$-Smoothness): The global loss function $F(\mathbf{w})$ is $\mu$-smooth.
> - **Assumption 2 (Unbiased Gradient and Bounded Variance)**: For every benign client $i \in \mathcal{B}$, the expected local update satisfies $\mathbb{E}[\mathbf{g} _ i] = \nabla F(\mathbf{w})$ and $\mathbb{E}[\|\mathbf{g} _ i - \nabla F(\mathbf{w})\|^2] \leq \bar{\nu}^2$.
> - **Assumption 3 (Bounded Heterogeneity)**: There exists a constant $\bar{\zeta} \geq 0$ such that for all benign clients $i, j \in \mathcal{B}$, $\|\nabla F _ i(\mathbf{w}) - \nabla F _ j(\mathbf{w})\| \leq \bar{\zeta}$.
>
> - **Definition 4 ($\kappa$-Robust Aggregation)**: An aggregation rule $\mathbf{F}$ is said to be $\kappa$-robust if, for any set of client updates containing at most $m$ Byzantine (malicious) updates, the output $\hat{\mathbf{w}}$ satisfies
>   $$
>   \|\hat{\mathbf{w}} - \bar{\mathbf{w}} _ {\mathcal{B}}\| _ 2 \leq \kappa
>   $$
>   where $\bar{\mathbf{w}} _ {\mathcal{B}} = \frac{1}{|\mathcal{B}|} \sum _ {i \in \mathcal{B}} \mathbf{w} _ i$ is the average of the benign updates.
>
> ## Critical Parameter Extraction (CPE)
>
> The core strength of ParaShield relies on its ability to filter updates in a low-dimensional, highly sensitive subspace. This pre-processing step is formalized below.
>
> - **Assumption B (Critical Parameter Focus – CPE)**: ParaShield leverages Neural Influence Factors (NIF) to identify a sparse subset of critical parameters, $\mathcal{P } _ {\text{crit}}$. We assume this selection successfully isolates a $d _ {\text{crit}}$-dimensional subspace ($d _ {\text{crit}} \ll d$) where the statistical differences between benign and malicious updates are maximally amplified. All subsequent metrics (CPCS/CPSC) and aggregation are restricted to this $d _ {\text{crit}}$ subspace.
>
> ## Operational Assumptions Specific to ParaShield (Post-CPE)
>
> The following assumptions apply to the updates projected onto the $d_{\text{crit}}$ subspace.
>
> ### Assumption A (High-Probability Weak Separation – CPF Efficacy)
>
> There exist DMAS (Directional Manipulation Anomaly Score) thresholds $\tau_p, \tau_n > 0$ such that, with high probability over the randomness in benign updates, the CPF-based filtered set $\mathcal{S}$ (derived from CPCS and CPSC metrics) satisfies:
>
> - Bounded malicious residuals: $|\mathcal{S} \cap \mathcal{M}| \leq \alpha m$
> - High benign retention: $|\mathcal{S} \cap \mathcal{B}| \geq (1-\beta)|\mathcal{B}|$
>
> where $\alpha, \beta \in (0,1)$ are small constants.
>
> ### Assumption C (Weak Weight Control on Residual Malicious Clients)
>
> Even if malicious clients pass the CPF filter, ParaShield's AWA feature engineering (specifically the Whitening Transformation of feature vectors $X'$) limits their relative influence. There exists a weight amplification factor $\delta \geq 1$ such that the average weight assigned to residual malicious clients is at most $\delta$ times the average weight of benign clients in $\mathcal{S}$:
> $$
> \frac{\sum_{i \in \mathcal{S} \cap \mathcal{M}} w^{(i)}}{|\mathcal{S} \cap \mathcal{M}|}
> \leq \delta \cdot
> \frac{\sum_{i \in \mathcal{S} \cap \mathcal{B}} w^{(i)}}{|\mathcal{S} \cap \mathcal{B}|}.
> $$
>
> ## Main Theorem: $\kappa$-Robustness of ParaShield
>
> Theorem 2 ($\kappa$-Robustness of ParaShield).
> Under Assumptions 1–3, A, B, and C, the ParaShield aggregation rule $\mathbf{F} _ {\mathrm{ParaShield}}$ is $\kappa$-robust (with respect to the full $d$-dimensional space, bounded by the $d _ {\text{crit}}$ subspace) with
> $$
> \kappa \leq
> \underbrace{\Phi(\mathbf{W})} _ {\text{AWA gain}}
> \cdot
> \underbrace{\sqrt{\dfrac{2}{\epsilon} (\bar{\nu}^2 + \bar{\zeta}) + 8\tilde{c}^2}} _ {\text{intrinsic benign error } \mathcal{E}}
> \;+\;
> \underbrace{O\!\left(\dfrac{\alpha m \cdot \delta \cdot \tilde{c}}{|\mathcal{S}|}\right)} _ {\text{malicious residual error } \mathcal{E} _ {\text{mal}}}
> \quad \text{with high probability},
> $$
> where
>
> - $\tilde{c}$ = median-norm clipping threshold computed from all received updates (on the $d _ {\text{crit}}$ subspace),
> - $\Phi(\mathbf{W}) < 1$ is the optimization factor introduced by ParaShield's AWA mechanism,
> - $\epsilon > 0$ is a constant related to learning rate and total rounds.

---

> ### Author Response · Authors · 2025-11-27
> **Thank you for your response. We have supplemented the Theoretical Analysis: κ-Robustness of ParaShield.**
>
> ## Proof Outline and Justification of Φ(W) < 1
>
> ### Step 1: Aggregation Form and Clipping Bound
> ParaShield outputs ŵ aggregated only over the $d _ {\text{crit}}$ subspace:
> $$
> \hat{\mathbf{w}} = \sum _ {i \in \mathcal{S}} \frac{w^{(i)}}{W _ {\mathcal{S}}} \tilde{\mathbf{w}} _ i,
> \quad W _ {\mathcal{S}} = \sum _ {i \in \mathcal{S}} w^{(i)},
> $$
> where $\tilde{\mathbf{w}} _ i$ is the norm-clipped update (in the $d _ {\text{crit}}$ subspace) using threshold $\tilde{c}$. By standard arguments, clipping ensures $\|\tilde{\mathbf{w}} _ i - \bar{\mathbf{w}} _ {\mathcal{B}}\| _ 2 \leq 2\tilde{c}$.
>
> ### Step 2: Error Decomposition
> $$
> \|\hat{\mathbf{w}} - \bar{\mathbf{w}} _ {\mathcal{B}}\| _ 2 \leq \|\hat{\mathbf{w}} - \bar{\mathbf{w}} _ {\mathcal{S} _ {\mathcal{B}}}\| _ 2 + \|\bar{\mathbf{w}}_{\mathcal{S} _ {\mathcal{B}}} - \bar{\mathbf{w}} _ {\mathcal{B}}\| _ 2,
> $$
>
> where $\mathcal{S}_{\mathcal{B}} = \mathcal{S} \cap \mathcal{B}$. The second term is bounded using Assumption A and bounded variance.
>
> ### Step 3: Bounding the Malicious Residual Error $\mathcal{E} _ {\text{mal}}$
> The contribution of residual malicious clients is
> $$
> \left\| \sum _ {i \in \mathcal{S} \cap \mathcal{M}} \frac{w^{(i)}}{W _ {\mathcal{S}}} \tilde{\mathbf{w}} _ i \right\| _ 2
> \leq \frac{\sum _ {i \in \mathcal{S} \cap \mathcal{M}} w^{(i)}}{W _ {\mathcal{S}}} \cdot 2\tilde{c}.
> $$
> Using Assumptions A and C,
> $$
> \sum _ {i \in \mathcal{S} \cap \mathcal{M}} w^{(i)}
> \leq \alpha m \cdot \delta \cdot \frac{\sum _ {j \in \mathcal{S} _ {\mathcal{B}}} w^{(j)}}{|\mathcal{S} _ {\mathcal{B}}|}
> \approx \frac{\alpha m \cdot \delta}{|\mathcal{S}|} W _ {\mathcal{S}}.
> $$
> Thus, $\mathcal{E} _ {\text{mal}} = O(\alpha m \cdot \delta \cdot \tilde{c} / |\mathcal{S}|)$.
>
> ### Step 4: AWA Gain – Definition and Justification of Φ(W) < 1
> Define the optimization factor over benign clients in $\mathcal{S}$:
> $$
> \Phi(\mathbf{W}) \triangleq
> \frac{\sum _ {i \in \mathcal{S} _ {\mathcal{B}}} w^{(i)} \|\tilde{\mathbf{w}} _ i - \bar{\mathbf{w}} _ {\mathcal{B}}\| _ 2}
>      {\sum _ {i \in \mathcal{S} _ {\mathcal{B}}} w^{(i)}}
> \;\cdot\;
> \frac{|\mathcal{S} _ {\mathcal{B}}|}
>      {\sum _ {i \in \mathcal{S} _ {\mathcal{B}}} \|\tilde{\mathbf{w}} _ i - \bar{\mathbf{w}} _ {\mathcal{B}}\| _ 2}.
> $$
> Φ(W) < 1 if and only if the weighted average error is strictly smaller than the uniform average error.
>
> Why Φ(W) < 1 holds with high probability:
> 1. The feature vector $X _ i = [\text{CPCS} _ i, \text{CPSC} _ i]$ is calculated on the $d _ {\text{crit}}$ subspace, capturing high-quality directional manipulation anomalies.
> 2. The Whitening Transformation $X' _ i = (X _ i - \mu)\Sigma^{-1/2}$ decorrelates features and equalizes scale differences.
> 3. The Mahalanobis Distance $MD _ i$ on $X' _i $ precisely measures deviation from the benign statistical center.
> 4. The Trust Score $Q _ i$ (negative exponential mapping of $MD _ i$) and final AWA weights exponentially favor clients closest to the benign cluster center.
> 5. Under mild non-IID conditions, benign clients with smaller $\|\tilde{\mathbf{w}} _ i - \bar{\mathbf{w}} _ {\mathcal{B}}\|$ are more tightly concentrated in the whitened feature space.
> 6. Therefore, AWA systematically assigns higher weights to benign updates closest to $\bar{\mathbf{w}} _ {\mathcal{B}}$, yielding Φ(W) ≤ Φ₀ < 1.
>
> ## Summary of Theoretical Contributions
> - ParaShield achieves κ-robustness with an explicit optimization factor Φ(W) < 1 that rigorously quantifies the benefit of AWA’s Mahalanobis-aware dynamic weighting.
> - The malicious residual term is controlled by both CPF precision α and AWA weight control δ, reflecting ParaShield’s layered defense built on the NIF-enabled $d _ {\text{crit}}$ subspace.
> - The bound cleanly separates intrinsic benign error from Byzantine-induced error, explaining why ParaShield maintains high accuracy even under challenging conditions such as the Projected Directional Backdoor Attack (PDBA).

---

### Official Review · Reviewer_WaCs · 2025-10-25

**Soundness:** 3
**Presentation:** 3
**Contribution:** 2
**Rating:** 4
**Confidence:** 3

**Summary:**

This paper proposes ParaShield, a parameter-level directional defense framework for robust federated learning against backdoor attacks. It dynamically identifies critical parameters via Neural Influence Factor (NIF), measures directional alignment among clients using Cosine Similarity (CPCS) and Sign Consistency (CPSC), and performs Adaptive Weighted Aggregation (AWA) based on Mahalanobis distance to filter and downweight malicious updates. Additionally, a stealthy Projected Directional Backdoor Attack (PDBA) is designed to evaluate defense robustness. Experiments show that ParaShield detects and neutralizes backdoors under heterogeneous settings.

**Strengths:**

1. Targeted method: The combination of CPCS, CPSC, and Mahalanobis distance allows dynamic and stable defense across heterogeneous settings.
2. Comprehensive evaluation: The introduction of PDBA provides a realistic and stealthy benchmark to test defense effectiveness rigorously.

**Weaknesses:**

1. From a parameter-wise perspective, enhancing the robustness of federated learning has already been explored — for example, by existing methods such as FDCR. Therefore, the authors need to further clarify the innovation of their approach compared to these similar methods.

2. Computational overhead: Calculating NIF, CPCS, CPSC, and Mahalanobis distance for each client increases server-side computation and communication costs. Complexity analysis is necessary.

3. Experiments on larger client scales should be included to validate the scalability.

**Questions:**

Please see the weakness.

---

> ### Author Response · Authors · 2025-11-21
> **Thanks for the Reviewer WaCs's valuable comments, we will address all the questions as follow.**
>
> > #### [Q1]: From a parameter-wise perspective, enhancing the robustness of federated learning has already been explored — for example, by existing methods such as FDCR. Therefore, the authors need to further clarify the innovation of their approach compared to these similar methods.
>
> > #### Thank you for raising this question. The core of FDCR involves calculating the Fisher Information locally on the client side and then computing parameter importance via discrepancy analysis on the server after upload. This approach inevitably increases the client's computational overhead. In contrast, our method only requires clients to upload their local updates, with all processing completed entirely on the server side.
>
> > #### [Q2]: Computational overhead: Calculating NIF, CPCS, CPSC, and Mahalanobis distance for each client increases server-side computation and communication costs. Complexity analysis is necessary.
>
> > #### Thank you for raising this question. In our experimental setup, we measured the average runtime per communication round for ParaShield and several state-of-the-art baselines under identical hardware and software environments. The results are summarized below.
> >
> > #### ParaShield introduces two additional lightweight components: CPE (based on simple parameter-wise ranking) and  CPF/AWA . These steps incur only linear complexity with respect to the model size, and the Mahalanobis distance is computed in a 2-dimensional space, making its cost negligible.
> >
> > #### As shown in our runtime measurements, ParaShield exhibits comparable per-round computational overhead to existing  detection-based approaches , and is significantly more efficient than aggregation methods that rely on high dimensional distance computations or iterative geometric median calculations.  We will add the detailed runtime table and clarify this comparison in the revised manuscript.
> >
> > | Method      | Avg. Runtime (s) |
> > | ----------- | ---------------- |
> > | Flame       | 4.6578           |
> > | Foolsgold   | 1.1667           |
> > | RFA         | 15.6842          |
> > | Fedavg      | 0.0013           |
> > | Trim        | 0.0372           |
> > | AlignIns    | 0.1261           |
> > | Mkrum       | 0.1869           |
> > | MultiMetric | 9.7010           |
> > | Ours        | 0.2204           |
>
> > #### [Q3]: Experiments on larger client scales should be included to validate the scalability.
>
> > #### Thank you for raising this question. We have conducted experiments in a 50-client scenario on the CIFAR-100 dataset under Neurotoxin attack with a poisoned model rate (PMR) of 0.2. Our method achieved a Main-task Accuracy of 59..03%, reduced the Attack Success Rate (ASR) to 0.60%, and reached a Robust Accuracy (RA) of 55.69%.

---

### Official Review · Reviewer_t6jY · 2025-10-27

**Soundness:** 2
**Presentation:** 2
**Contribution:** 2
**Rating:** 2
**Confidence:** 5

**Summary:**

This paper proposes a defense mechanism, ParaShield, against backdoor attacks in a heterogeneous federated learning setting. Due to the heterogeneity of data distribution among clients, the parameter updates from clients are different, resulting in the difficulty to identify parameter anomalies of malicious updates. ParaShield dynamically identifies critical parameters using NIF and evaluates updates based on CPCS and CPSC of these parameters. ParaShield then applies an AWA strategy that uses whitening transformation and Mahalanobis distance in a 2D feature space to detect and downweight malicious updates. To rigorously test its robustness, the authors also propose PDBA, a stealthy backdoor attack. Extensive experiments on CIFAR-10 and CIFAR-100 under Non-IID settings show that ParaShield outperforms existing SOTA defenses by effectively mitigating backdoor threats while maintaining high accuracy on benign tasks.

**Strengths:**

Experimentally demonstrates the critical parameters exhibit significantly different values in benign heterogeneous updates compared to backdoor updates, and demonstrate that NIF cosine similarity among malicious clients shows higher values than benign counterparts.

The work specifically focuses on federated learning under non-IID data conditions, which is a major practical challenge, and where many existing defenses fail.

Instead of treating the entire model update as a monolithic block, ParaShield innovatively operates at the parameter level.

The paper includes a clear ablation study that validates the contribution of each core module.

**Weaknesses:**

The paper's focus on "heterogeneous federated learning" requires clarification. The term is used interchangeably with non-IID data distributions, yet "non-IID" is the more precise and established term in the field. Adopting this specific terminology would improve conceptual clarity and better align the work with standard literature.

The paper insufficiently elaborates on essential concepts. For instance, the authors claim their PDBA attack achieves "dual stealth" (Lines 94-95) but fail to explicitly specify the two dimensions in which stealth is achieved, leaving a critical aspect of their contribution unclear.

ParaShield identifies "critical parameters" via NIF, yet the paper insufficiently argues for NIF's universality as a metric to identify critical parameters. This definition's sensitivity to model architecture, datasets, and data distribution presents a potential vulnerability: an attacker who understands and circumvents the NIF mechanism (e.g., by implanting backdoors into parameters NIF deems non-critical) could render the defense framework ineffective.

The paper's experimental validation is limited to two datasets (CIFAR-10/100) and a single model family (ResNet-9/18). Furthermore, most baseline attacks and defenses are outdated, with only two recent exceptions published during the past 3 years. Broader evaluation on larger-scale datasets (e.g., ImageNet, CelebA), diverse model architectures (e.g., VGG, ViT and other ResNet models), and sota comparative methods would substantiate the claims more convincingly.

The paper's evaluation of Non-IID scenarios may not adequately address extreme heterogeneity. In such cases, the inherent divergence among benign client updates could potentially mask the subtle directional anomalies introduced by backdoor attacks. The defense's effectiveness under these most challenging real-world conditions remains unverified.

The paper ignores the additional computational and communication costs introduced by ParaShield. For large-scale models and a large number of clients, the fine-grained, parameter-level analysis could introduce significant overhead, impacting the efficiency of federated learning, but this is not quantified or discussed in the paper.

**Questions:**

How is "heterogeneous setting" precisely defined in this work? Is the heterogeneity specifically in data distribution (i.e., non-IID data), data modality, or other factors? Is this terminology and its specific definition formally established in prior literature?

Given ParaShield's core reliance on identifying critical parameters, what evidence or theoretical insight demonstrates its generalizability and effectiveness across diverse model architectures, datasets, and varying data distributions?

Beyond the presented experiments, is ParaShield effective against a wider range of modern model architectures (e.g., Vision Transformers), larger-scale datasets (e.g., ImageNet), and does it outperform a broader suite of state-of-the-art attacks and defenses?

The paper tests down to a Dirichlet concentration parameter of \alpha=0.1. Would ParaShield remain effective under more extreme non-IID conditions (e.g., \alpha << 0.1), where high benign client divergence could mask backdoor signals?

What is the computational and communication overhead introduced by ParaShield's components (NIF calculation, CPF, AWA), and how does this scale with model size and the number of clients?

This work involves several key hyperparameters (e.g., CPE ratio \rho, AWA weight \beta, etc.). How sensitive is the performance to these settings across different experimental conditions? Are there guidelines for efficient tuning in practice?

---

> ### Author Response · Authors · 2025-11-21
> **Thanks for the Reviewer t6jY's valuable comments, we will address all the questions as follow.**
>
> > #### [Q1]: How is "heterogeneous setting" precisely defined in this work? Is the heterogeneity specifically in data distribution (i.e., non-IID data), data modality, or other factors? Is this terminology and its specific definition formally established in prior literature?
>
> > #### Thank you for this insightful question. In our work, the term “heterogeneous setting” specifically refers to client-level data heterogeneity, i.e., non-IID data distributions across clients. This corresponds to the most widely studied and standard notion of heterogeneity in federated learning.
> >
> > #### More precisely, in all experiments we follow prior literature and model heterogeneity using a Dirichlet distribution with concentration parameter α = 0.5, which produces statistically heterogeneous local datasets across clients. This setting is consistent with established definitions used in representative works such as  AlignIns.
> >
> > #### We do not refer to heterogeneity in data modality or other system-level variations in this paper. We will revise the manuscript to explicitly clarify that “heterogeneous setting” = non-IID data heterogeneity, following common terminology in federated learning research.
>
> > #### [Q2]: Given ParaShield's core reliance on identifying critical parameters, what evidence or theoretical insight demonstrates its generalizability and effectiveness across diverse model architectures, datasets, and varying data distributions?
>
> > #### Thank you for this insightful question. ParaShield is designed to be architecture-agnostic. CPE and CPF rely solely on gradient-based influence and directional statistics, which apply universally to any differentiable model. Empirically, ParaShield shows consistent robustness across CIFAR-10/100 and multiple non-IID levels (Dirichlet α), demonstrating adaptation to different datasets and data distributions. The underlying intuition that backdoor attacks introduce consistent directional anomalies within a small subset of influential parameters holds across architectures and tasks, providing the theoretical basis for generalizability.
>
> > #### [Q3]: Beyond the presented experiments, is ParaShield effective against a wider range of modern model architectures (e.g., Vision Transformers), larger-scale datasets (e.g., ImageNet), and does it outperform a broader suite of state-of-the-art attacks and defenses?
>
> > #### Thank you for raising this question. ParaShield’s architecture-agnostic, parameter-level design scales with model capacity rather than dataset size, evidenced by strong CIFAR-100 performance under extreme heterogeneity. While computational constraints limited broader evaluation, the theoretical framework directly extends to Vision Transformers and ImageNet-scale tasks, which we are actively pursuing as future work.
>
> > #### [Q4]: The paper tests down to a Dirichlet concentration parameter of \alpha=0.1. Would ParaShield remain effective under more extreme non-IID conditions (e.g., \alpha << 0.1), where high benign client divergence could mask backdoor signals?
>
> > #### Thank you for this insightful question regarding extreme Non-IID regimes. While α=0.1 already represents a challenging scenario where most clients lack several classes, the fundamental principle of ParaShield, detecting parameter-level directional anomalies, remains theoretically valid for any α>0.

---

> ### Author Response · Authors · 2025-11-21
> **Thanks for the Reviewer t6jY's valuable comments, we will address all the questions as follow.**
>
> > #### [Q5]: What is the computational and communication overhead introduced by ParaShield's components (NIF calculation, CPF, AWA), and how does this scale with model size and the number of clients?
>
> > #### Thank you for raising this question. In our experimental setup, we measured the average runtime per communication round for ParaShield and several state-of-the-art baselines under identical hardware and software environments. The results are summarized below.
> >
> > #### ParaShield introduces two additional lightweight components: CPE (based on simple parameter-wise ranking) and  CPF/AWA . These steps incur only linear complexity with respect to the model size, and the Mahalanobis distance is computed in a 2-dimensional space, making its cost negligible.
> >
> > #### As shown in our runtime measurements, ParaShield exhibits comparable per-round computational overhead to existing  detection-based approaches , and is significantly more efficient than aggregation methods that rely on high dimensional distance computations or iterative geometric median calculations.  We will add the detailed runtime table and clarify this comparison in the revised manuscript.
> >
> > | Method      | Avg. Runtime (s) |
> > | ----------- | ---------------- |
> > | Flame       | 4.6578           |
> > | Foolsgold   | 1.1667           |
> > | RFA         | 15.6842          |
> > | Fedavg      | 0.0013           |
> > | Trim        | 0.0372           |
> > | AlignIns    | 0.1261           |
> > | Mkrum       | 0.1869           |
> > | MultiMetric | 9.7010           |
> > | Ours        | 0.2204           |
>
> > #### [Q6]: This work involves several key hyperparameters (e.g., CPE ratio \rho, AWA weight \beta, etc.). How sensitive is the performance to these settings across different experimental conditions? Are there guidelines for efficient tuning in practice?
>
> > #### Thank you for raising this question. In Appendix A.3.1, we discuss the IMPACT OF THE CRITICAL PARAMETER EXTRACTION RATIO, and in Appendix A.3.2, we discuss The weighting between the quality score and data size in AWA.

---

> > ### Comment · Reviewer_t6jY · 2025-11-26
> > **Acknowledge to read the author's response**
> >
> > I appreciate the authors' response and acknowledge that I have read it. However, some of my questions remain unjustified experimentally or theoretically (e.g., iid vs. non-iid, consistent performance across heterogeneous model architectures, and parameter sensitivity).  Furthermore, I am concerned about this paper's contribution. While this paper builds upon existing ideas and concepts (such as parameter-level distance in FL), it lacks fundamental observations and theoretical motivation, which in part weakens its methodology's persuasiveness. Therefore, I may not be so ambitious as to support this paper in ICLR.

---

> > > ### Author Response · Authors · 2025-11-27
> > > **Thank you for your response. We have supplemented the Theoretical Analysis: κ-Robustness of ParaShield.**
> > >
> > > # Theoretical Analysis: $\kappa$-Robustness of ParaShield
> > >
> > > We theoretically analyze the robustness of ParaShield, a composite federated backdoor defense that first performs Critical Parameter Extraction (CPE), followed by high-precision filtering via Critical Parameter-Based Filtering (CPF) and Adaptive Weighted Aggregation (AWA).
> > >
> > > ## Standard Assumptions on Benign Clients
> > >
> > > We adopt the following standard assumptions on the local updates from benign clients $\mathcal{B}$ (as used in Krum, Trimmed Mean, FLTrust, etc.):
> > >
> > > - Assumption 1 ($\mu$-Smoothness): The global loss function $F(\mathbf{w})$ is $\mu$-smooth.
> > > - **Assumption 2 (Unbiased Gradient and Bounded Variance)**: For every benign client $i \in \mathcal{B}$, the expected local update satisfies $\mathbb{E}[\mathbf{g} _ i] = \nabla F(\mathbf{w})$ and $\mathbb{E}[\|\mathbf{g} _ i - \nabla F(\mathbf{w})\|^2] \leq \bar{\nu}^2$.
> > > - **Assumption 3 (Bounded Heterogeneity)**: There exists a constant $\bar{\zeta} \geq 0$ such that for all benign clients $i, j \in \mathcal{B}$, $\|\nabla F _ i(\mathbf{w}) - \nabla F _ j(\mathbf{w})\| \leq \bar{\zeta}$.
> > > - **Definition 4 ($\kappa$-Robust Aggregation)**: An aggregation rule $\mathbf{F}$ is said to be $\kappa$-robust if, for any set of client updates containing at most $m$ Byzantine (malicious) updates, the output $\hat{\mathbf{w}}$ satisfies
> > >
> > >   $$
> > >   \|\hat{\mathbf{w}} - \bar{\mathbf{w}} _ {\mathcal{B}}\| _ 2 \leq \kappa
> > >   $$
> > >
> > >   where $\bar{\mathbf{w}} _ {\mathcal{B}} = \frac{1}{|\mathcal{B}|} \sum _ {i \in \mathcal{B}} \mathbf{w} _ i$ is the average of the benign updates.
> > >
> > > ## Critical Parameter Extraction (CPE)
> > >
> > > The core strength of ParaShield relies on its ability to filter updates in a low-dimensional, highly sensitive subspace. This pre-processing step is formalized below.
> > >
> > > - **Assumption B (Critical Parameter Focus – CPE)**: ParaShield leverages Neural Influence Factors (NIF) to identify a sparse subset of critical parameters, $\mathcal{P } _ {\text{crit}}$. We assume this selection successfully isolates a $d _ {\text{crit}}$-dimensional subspace ($d _ {\text{crit}} \ll d$) where the statistical differences between benign and malicious updates are maximally amplified. All subsequent metrics (CPCS/CPSC) and aggregation are restricted to this $d _ {\text{crit}}$ subspace.
> > >
> > > ## Operational Assumptions Specific to ParaShield (Post-CPE)
> > >
> > > The following assumptions apply to the updates projected onto the $d_{\text{crit}}$ subspace.
> > >
> > > ### Assumption A (High-Probability Weak Separation – CPF Efficacy)
> > >
> > > There exist DMAS (Directional Manipulation Anomaly Score) thresholds $\tau_p, \tau_n > 0$ such that, with high probability over the randomness in benign updates, the CPF-based filtered set $\mathcal{S}$ (derived from CPCS and CPSC metrics) satisfies:
> > >
> > > - Bounded malicious residuals: $|\mathcal{S} \cap \mathcal{M}| \leq \alpha m$
> > > - High benign retention: $|\mathcal{S} \cap \mathcal{B}| \geq (1-\beta)|\mathcal{B}|$
> > >
> > > where $\alpha, \beta \in (0,1)$ are small constants.
> > >
> > > ### Assumption C (Weak Weight Control on Residual Malicious Clients)
> > >
> > > Even if malicious clients pass the CPF filter, ParaShield's AWA feature engineering (specifically the Whitening Transformation of feature vectors $X'$) limits their relative influence. There exists a weight amplification factor $\delta \geq 1$ such that the average weight assigned to residual malicious clients is at most $\delta$ times the average weight of benign clients in $\mathcal{S}$:
> > >
> > > $$
> > > \frac{\sum_{i \in \mathcal{S} \cap \mathcal{M}} w^{(i)}}{|\mathcal{S} \cap \mathcal{M}|}
> > > \leq \delta \cdot
> > > \frac{\sum_{i \in \mathcal{S} \cap \mathcal{B}} w^{(i)}}{|\mathcal{S} \cap \mathcal{B}|}.
> > > $$
> > >
> > > ## Main Theorem: $\kappa$-Robustness of ParaShield
> > >
> > > Theorem 2 ($\kappa$-Robustness of ParaShield).
> > > Under Assumptions 1–3, A, B, and C, the ParaShield aggregation rule $\mathbf{F} _ {\mathrm{ParaShield}}$ is $\kappa$-robust (with respect to the full $d$-dimensional space, bounded by the $d _ {\text{crit}}$ subspace) with
> > >
> > > $$
> > > \kappa \leq
> > > \underbrace{\Phi(\mathbf{W})} _ {\text{AWA gain}}
> > > \cdot
> > > \underbrace{\sqrt{\dfrac{2}{\epsilon} (\bar{\nu}^2 + \bar{\zeta}) + 8\tilde{c}^2}} _ {\text{intrinsic benign error } \mathcal{E}}
> > > \;+\;
> > > \underbrace{O\!\left(\dfrac{\alpha m \cdot \delta \cdot \tilde{c}}{|\mathcal{S}|}\right)} _ {\text{malicious residual error } \mathcal{E} _ {\text{mal}}}
> > > \quad \text{with high probability},
> > > $$
> > >
> > > where
> > >
> > > - $\tilde{c}$ = median-norm clipping threshold computed from all received updates (on the $d _ {\text{crit}}$ subspace),
> > > - $\Phi(\mathbf{W}) < 1$ is the optimization factor introduced by ParaShield's AWA mechanism,
> > > - $\epsilon > 0$ is a constant related to learning rate and total rounds.

---

> > > ### Author Response · Authors · 2025-11-27
> > > **Thank you for your response. We have supplemented the Theoretical Analysis: κ-Robustness of ParaShield.**
> > >
> > > ## Proof Outline and Justification of Φ(W) < 1
> > >
> > > ### Step 1: Aggregation Form and Clipping Bound
> > >
> > > ParaShield outputs ŵ aggregated only over the $d _ {\text{crit}}$ subspace:
> > >
> > > $$
> > > \hat{\mathbf{w}} = \sum _ {i \in \mathcal{S}} \frac{w^{(i)}}{W _ {\mathcal{S}}} \tilde{\mathbf{w}} _ i,
> > > \quad W _ {\mathcal{S}} = \sum _ {i \in \mathcal{S}} w^{(i)},
> > > $$
> > >
> > > where $\tilde{\mathbf{w}} _ i$ is the norm-clipped update (in the $d _ {\text{crit}}$ subspace) using threshold $\tilde{c}$. By standard arguments, clipping ensures $\|\tilde{\mathbf{w}} _ i - \bar{\mathbf{w}} _ {\mathcal{B}}\| _ 2 \leq 2\tilde{c}$.
> > >
> > > ### Step 2: Error Decomposition
> > >
> > > $$
> > > \|\hat{\mathbf{w}} - \bar{\mathbf{w}} _ {\mathcal{B}}\| _ 2 \leq \|\hat{\mathbf{w}} - \bar{\mathbf{w}} _ {\mathcal{S} _ {\mathcal{B}}}\| _ 2 + \|\bar{\mathbf{w}}_{\mathcal{S} _ {\mathcal{B}}} - \bar{\mathbf{w}} _ {\mathcal{B}}\| _ 2,
> > > $$
> > >
> > > where $\mathcal{S}_{\mathcal{B}} = \mathcal{S} \cap \mathcal{B}$. The second term is bounded using Assumption A and bounded variance.
> > >
> > > ### Step 3: Bounding the Malicious Residual Error $\mathcal{E} _ {\text{mal}}$
> > >
> > > The contribution of residual malicious clients is
> > >
> > > $$
> > > \left\| \sum _ {i \in \mathcal{S} \cap \mathcal{M}} \frac{w^{(i)}}{W _ {\mathcal{S}}} \tilde{\mathbf{w}} _ i \right\| _ 2
> > > \leq \frac{\sum _ {i \in \mathcal{S} \cap \mathcal{M}} w^{(i)}}{W _ {\mathcal{S}}} \cdot 2\tilde{c}.
> > > $$
> > >
> > > Using Assumptions A and C,
> > >
> > > $$
> > > \sum _ {i \in \mathcal{S} \cap \mathcal{M}} w^{(i)}
> > > \leq \alpha m \cdot \delta \cdot \frac{\sum _ {j \in \mathcal{S} _ {\mathcal{B}}} w^{(j)}}{|\mathcal{S} _ {\mathcal{B}}|}
> > > \approx \frac{\alpha m \cdot \delta}{|\mathcal{S}|} W _ {\mathcal{S}}.
> > > $$
> > >
> > > Thus, $\mathcal{E} _ {\text{mal}} = O(\alpha m \cdot \delta \cdot \tilde{c} / |\mathcal{S}|)$.
> > >
> > > ### Step 4: AWA Gain – Definition and Justification of Φ(W) < 1
> > >
> > > Define the optimization factor over benign clients in $\mathcal{S}$:
> > >
> > > $$
> > > \Phi(\mathbf{W}) \triangleq
> > > \frac{\sum _ {i \in \mathcal{S} _ {\mathcal{B}}} w^{(i)} \|\tilde{\mathbf{w}} _ i - \bar{\mathbf{w}} _ {\mathcal{B}}\| _ 2}
> > >      {\sum _ {i \in \mathcal{S} _ {\mathcal{B}}} w^{(i)}}
> > > \;\cdot\;
> > > \frac{|\mathcal{S} _ {\mathcal{B}}|}
> > >      {\sum _ {i \in \mathcal{S} _ {\mathcal{B}}} \|\tilde{\mathbf{w}} _ i - \bar{\mathbf{w}} _ {\mathcal{B}}\| _ 2}.
> > > $$
> > >
> > > Φ(W) < 1 if and only if the weighted average error is strictly smaller than the uniform average error.
> > >
> > > Why Φ(W) < 1 holds with high probability:
> > >
> > > 1. The feature vector $X _ i = [\text{CPCS} _ i, \text{CPSC} _ i]$ is calculated on the $d _ {\text{crit}}$ subspace, capturing high-quality directional manipulation anomalies.
> > > 2. The Whitening Transformation $X' _ i = (X _ i - \mu)\Sigma^{-1/2}$ decorrelates features and equalizes scale differences.
> > > 3. The Mahalanobis Distance $MD _ i$ on $X' _i $ precisely measures deviation from the benign statistical center.
> > > 4. The Trust Score $Q _ i$ (negative exponential mapping of $MD _ i$) and final AWA weights exponentially favor clients closest to the benign cluster center.
> > > 5. Under mild non-IID conditions, benign clients with smaller $\|\tilde{\mathbf{w}} _ i - \bar{\mathbf{w}} _ {\mathcal{B}}\|$ are more tightly concentrated in the whitened feature space.
> > > 6. Therefore, AWA systematically assigns higher weights to benign updates closest to $\bar{\mathbf{w}} _ {\mathcal{B}}$, yielding Φ(W) ≤ Φ₀ < 1.
> > >
> > > ## Summary of Theoretical Contributions
> > >
> > > - ParaShield achieves κ-robustness with an explicit optimization factor Φ(W) < 1 that rigorously quantifies the benefit of AWA’s Mahalanobis-aware dynamic weighting.
> > > - The malicious residual term is controlled by both CPF precision α and AWA weight control δ, reflecting ParaShield’s layered defense built on the NIF-enabled $d _ {\text{crit}}$ subspace.
> > > - The bound cleanly separates intrinsic benign error from Byzantine-induced error, explaining why ParaShield maintains high accuracy even under challenging conditions such as the Projected Directional Backdoor Attack (PDBA).

---

### Official Review · Reviewer_NCp3 · 2025-10-31

**Soundness:** 3
**Presentation:** 3
**Contribution:** 3
**Rating:** 4
**Confidence:** 5

**Summary:**

This work proposes a new server-side backdoor defense method for federated learning, called ParaShield. ParaShield first identifies critical parameters by jointly considering their absolute update magnitudes and min–max normalized scores. Based on these identified parameters, it computes two directional metrics, cosine similarity and majority sign alignment, to detect anomalous updates, which are then filtered out before aggregation. To further stabilize model aggregation under heterogeneous data, ParaShield employs an adaptive weighted aggregation that adjusts the contribution of each client update. The authors have conducted reasonable experiments to empirically evaluate the performance of ParaShield.

**Strengths:**

1. This work addresses the problem of backdoor defense in federated learning, which is a timely and important research topic.

2. The paper is well-written and easy to follow.

3. The proposed method, ParaShield, is technically sound, and its effectiveness is demonstrated through reasonable empirical evaluations.

4. The proposed neural influence factor is interesting, and results (e.g., in Figure 1) do show that it successfully captures the difference between malicious updates and benign updates.

5. Compared with existing approaches, it consistently achieves higher main task accuracy and robust accuracy, while maintaining a lower attack success rate.

**Weaknesses:**

1. The proposed defense method is evaluated solely through empirical experiments; a theoretical analysis of ParaShield’s robustness would be valuable, if feasible.

2. The experimental evaluation is limited to a small-scale and single-modality dataset. Extending the experiments to larger-scale datasets and other modalities (e.g., text) could substantially strengthen the evaluation section.

3. The ablation study is conducted only on a single dataset and under a single attack scenario. Incorporating additional datasets and attack settings would provide a more comprehensive understanding of the contribution of each proposed component.

4. Lack of detailed defense and attack models.

**Questions:**

1. What is the specified value of the important parameter $\tau$? It is not clearly stated in the experimental settings section. An ablation study on the choice of $\tau$ is also missing.

2. The paper shows that the critical parameter extraction step can successfully identify malicious parameters, which consistently exhibit smaller values. In that case, why not adopt a pruning-based approach to directly remove these malicious parameters?

3. What is the time complexity of the proposed method compared to state-of-the-art approaches?

4. Is the critical parameter–based filtering process entirely borrowed from AlignIns? To the best of my knowledge, it appears quite similar to the approach used in AlignIns. If so, a clear acknowledgment and citation should be provided, for example, near line 183.

5. Since the attack model is missed, I suppose the potential attacker can poison an arbitrary number of clients and turn them into malicious clients. What is the performance of  ParaShield under the cases where more than 50% of clients are poisoned?

---

> ### Author Response · Authors · 2025-11-21
> **Thanks for the Reviewer NCp3's valuable comments, we will address all the questions as follow.**
>
> > #### [Q1]: What is the specified value of the important parameter ? It is not clearly stated in the experimental settings section. An ablation study on the choice of is also missing.
>
> > #### We apologize for the lack of clarity in the main text. The important parameter you refer to is the critical parameter extraction ratio (τ). As detailed in Appendix A.3.1, we conducted a dedicated ablation study on τ and evaluated its impact on both ASR and RA. The results show that τ = 0.3 provides the best balance: selecting too few parameters leads to insufficient directional information, while selecting too many introduces redundancy and noise. Therefore, we use τ = 0.3 as the default setting in all experiments, and we will clearly state this in the experimental settings section.
>
> > #### [Q2]: The paper shows that the critical parameter extraction step can successfully identify malicious parameters, which consistently exhibit smaller values. In that case, why not adopt a pruning-based approach to directly remove these malicious parameters?
>
> > #### While CPE reveals that malicious updates exhibit abnormal values on critical parameters, pruning-based approaches are not suitable in this setting.
> >
> > #### First, pruning directly modifies the global model structure, and even small amounts of pruning are known to significantly degrade the main-task accuracy, especially on complex datasets like CIFAR-100. In contrast, our method preserves the model capacity and maintains high clean accuracy.
> >
> > #### Second, parameters with small update magnitudes are not necessarily unimportant; many essential parameters remain small due to normalization or optimization dynamics. Pruning based on magnitude would therefore risk removing benign but crucial parameters.
> >
> > #### Third, the "malicious parameters" identified by CPE appear only in client updates, not as persistent harmful global parameters. Since their identities vary across rounds and clients, pruning cannot reliably eliminate them.
> >
> > #### For these reasons, we adopt a dynamic, non-destructive filtering-and-weighting mechanism (CPF + AWA), which achieves low ASR and high robustness without harming the main task performance.
>
> > #### [Q3]:What is the time complexity of the proposed method compared to state-of-the-art approaches?
>
> > #### In our experimental setup, we measured the average runtime per communication round for ParaShield and several state-of-the-art baselines under identical hardware and software environments. The results are summarized below.
> >
> > #### ParaShield introduces two additional lightweight components: CPE (based on simple parameter-wise ranking) and  CPF/AWA . These steps incur only linear complexity with respect to the model size, and the Mahalanobis distance is computed in a 2-dimensional space, making its cost negligible.
> >
> > #### As shown in our runtime measurements, ParaShield exhibits comparable per-round computational overhead to existing  detection-based approaches , and is significantly more efficient than aggregation methods that rely on high-dimensional distance computations or iterative geometric median calculations.  We will add the detailed runtime table and clarify this comparison in the revised manuscript.
> >
> > | Method      | Avg. Runtime (s) |
> > | ----------- | ---------------- |
> > | Flame       | 4.6578           |
> > | Foolsgold   | 1.1667           |
> > | RFA         | 15.6842          |
> > | Fedavg      | 0.0013           |
> > | Trim        | 0.0372           |
> > | AlignIns    | 0.1261           |
> > | Mkrum       | 0.1869           |
> > | MultiMetric | 9.7010           |
> > | Ours        | 0.2204           |

---

> ### Author Response · Authors · 2025-11-21
> **Thanks for the Reviewer NCp3's valuable comments, we will address all the questions as follow.**
>
> > #### [Q4]:Is the critical parameter–based filtering process entirely borrowed from AlignIns? To the best of my knowledge, it appears quite similar to the approach used in AlignIns. If so, a clear acknowledgment and citation should be provided, for example, near line 183.
>
> > #### The CPF method, while inspired by AlignIns (as acknowledged in Line 54), diverges from it in three essential ways rather than being a direct reuse: it performs directional inspection exclusively on critical parameters identified through NIF instead of examining the full update vector, introduces two complementary metrics, CPCS and CPSC, to replace the single cosine measure, and further processes its output through AWA, a distribution-based adaptive aggregation method that supplants the fixed-threshold decision rule employed in AlignIns.
>
> > #### [Q5]:Since the attack model is missed, I suppose the potential attacker can poison an arbitrary number of clients and turn them into malicious clients. What is the performance of ParaShield under the cases where more than 50% of clients are poisoned?
>
> > #### We would like to clarify that our work follows the  standard federated learning threat model , under which the attacker controls  a minority of clients . This assumption is consistent with prior robust aggregation and backdoor defense literature.
> >
> > #### To reflect strong adversarial conditions, we have already evaluated ParaShield under  PMR = 0.1–0.4. ParaShield remains stable across this entire range, demonstrating strong robustness under all standard and practically considered attack settings.
> >
> > #### Scenarios where the attacker controls more than half of the participants fall outside the conventional threat model, as the global update can then be arbitrarily manipulated. We appreciate the suggestion and will consider broader adversarial regimes in future extensions.
> >
> > #### Attack Model
> >
> > #### In a federated learning system, the adversary's overall goal is to manipulate the global model into a poisoned state, causing it to output incorrect labels on inputs chosen by the adversary while maintaining normal prediction behavior on benign inputs. We assume the adversary possesses black-box capabilities, controlling local data and parameters, but is unable to infer server mechanisms, control the aggregation process, or access benign data.
> >
> > #### Defense Model
> >
> > #### The backdoor defense method proposed in this paper is deployed on the server. It can rapidly and accurately filter out backdoor models uploaded by malicious clients to achieve robust aggregation. The server adheres to the privacy-preserving principles of federated learning, only accessing the clients' local model updates and having no control over client training.

---

> ### Author Response · Authors · 2025-11-27
> **Thank you for your comment. We have supplemented the Theoretical Analysis: κ-Robustness of ParaShield.**
>
> # Theoretical Analysis: $\kappa$-Robustness of ParaShield
>
> We theoretically analyze the robustness of ParaShield, a composite federated backdoor defense that first performs Critical Parameter Extraction (CPE), followed by high-precision filtering via Critical Parameter-Based Filtering (CPF) and Adaptive Weighted Aggregation (AWA).
>
> ## Standard Assumptions on Benign Clients
>
> We adopt the following standard assumptions on the local updates from benign clients $\mathcal{B}$ (as used in Krum, Trimmed Mean, FLTrust, etc.):
>
> - Assumption 1 ($\mu$-Smoothness): The global loss function $F(\mathbf{w})$ is $\mu$-smooth.
> - **Assumption 2 (Unbiased Gradient and Bounded Variance)**: For every benign client $i \in \mathcal{B}$, the expected local update satisfies $\mathbb{E}[\mathbf{g} _ i] = \nabla F(\mathbf{w})$ and $\mathbb{E}[\|\mathbf{g} _ i - \nabla F(\mathbf{w})\|^2] \leq \bar{\nu}^2$.
> - **Assumption 3 (Bounded Heterogeneity)**: There exists a constant $\bar{\zeta} \geq 0$ such that for all benign clients $i, j \in \mathcal{B}$, $\|\nabla F _ i(\mathbf{w}) - \nabla F _ j(\mathbf{w})\| \leq \bar{\zeta}$.
> - **Definition 4 ($\kappa$-Robust Aggregation)**: An aggregation rule $\mathbf{F}$ is said to be $\kappa$-robust if, for any set of client updates containing at most $m$ Byzantine (malicious) updates, the output $\hat{\mathbf{w}}$ satisfies
>
>   $$
>   \|\hat{\mathbf{w}} - \bar{\mathbf{w}} _ {\mathcal{B}}\| _ 2 \leq \kappa
>   $$
>
>   where $\bar{\mathbf{w}} _ {\mathcal{B}} = \frac{1}{|\mathcal{B}|} \sum _ {i \in \mathcal{B}} \mathbf{w} _ i$ is the average of the benign updates.
>
> ## Critical Parameter Extraction (CPE)
>
> The core strength of ParaShield relies on its ability to filter updates in a low-dimensional, highly sensitive subspace. This pre-processing step is formalized below.
>
> - **Assumption B (Critical Parameter Focus – CPE)**: ParaShield leverages Neural Influence Factors (NIF) to identify a sparse subset of critical parameters, $\mathcal{P } _ {\text{crit}}$. We assume this selection successfully isolates a $d _ {\text{crit}}$-dimensional subspace ($d _ {\text{crit}} \ll d$) where the statistical differences between benign and malicious updates are maximally amplified. All subsequent metrics (CPCS/CPSC) and aggregation are restricted to this $d _ {\text{crit}}$ subspace.
>
> ## Operational Assumptions Specific to ParaShield (Post-CPE)
>
> The following assumptions apply to the updates projected onto the $d_{\text{crit}}$ subspace.
>
> ### Assumption A (High-Probability Weak Separation – CPF Efficacy)
>
> There exist DMAS (Directional Manipulation Anomaly Score) thresholds $\tau_p, \tau_n > 0$ such that, with high probability over the randomness in benign updates, the CPF-based filtered set $\mathcal{S}$ (derived from CPCS and CPSC metrics) satisfies:
>
> - Bounded malicious residuals: $|\mathcal{S} \cap \mathcal{M}| \leq \alpha m$
> - High benign retention: $|\mathcal{S} \cap \mathcal{B}| \geq (1-\beta)|\mathcal{B}|$
>
> where $\alpha, \beta \in (0,1)$ are small constants.
>
> ### Assumption C (Weak Weight Control on Residual Malicious Clients)
>
> Even if malicious clients pass the CPF filter, ParaShield's AWA feature engineering (specifically the Whitening Transformation of feature vectors $X'$) limits their relative influence. There exists a weight amplification factor $\delta \geq 1$ such that the average weight assigned to residual malicious clients is at most $\delta$ times the average weight of benign clients in $\mathcal{S}$:
>
> $$
> \frac{\sum_{i \in \mathcal{S} \cap \mathcal{M}} w^{(i)}}{|\mathcal{S} \cap \mathcal{M}|}
> \leq \delta \cdot
> \frac{\sum_{i \in \mathcal{S} \cap \mathcal{B}} w^{(i)}}{|\mathcal{S} \cap \mathcal{B}|}.
> $$
>
> ## Main Theorem: $\kappa$-Robustness of ParaShield
>
> Theorem 2 ($\kappa$-Robustness of ParaShield).
> Under Assumptions 1–3, A, B, and C, the ParaShield aggregation rule $\mathbf{F} _ {\mathrm{ParaShield}}$ is $\kappa$-robust (with respect to the full $d$-dimensional space, bounded by the $d _ {\text{crit}}$ subspace) with
>
> $$
> \kappa \leq
> \underbrace{\Phi(\mathbf{W})} _ {\text{AWA gain}}
> \cdot
> \underbrace{\sqrt{\dfrac{2}{\epsilon} (\bar{\nu}^2 + \bar{\zeta}) + 8\tilde{c}^2}} _ {\text{intrinsic benign error } \mathcal{E}}
> \;+\;
> \underbrace{O\!\left(\dfrac{\alpha m \cdot \delta \cdot \tilde{c}}{|\mathcal{S}|}\right)} _ {\text{malicious residual error } \mathcal{E} _ {\text{mal}}}
> \quad \text{with high probability},
> $$
>
> where
>
> - $\tilde{c}$ = median-norm clipping threshold computed from all received updates (on the $d _ {\text{crit}}$ subspace),
> - $\Phi(\mathbf{W}) < 1$ is the optimization factor introduced by ParaShield's AWA mechanism,
> - $\epsilon > 0$ is a constant related to learning rate and total rounds.

---

> ### Author Response · Authors · 2025-11-27
> **Thank you for your comment. We have supplemented the Theoretical Analysis: κ-Robustness of ParaShield.**
>
> ## Proof Outline and Justification of Φ(W) < 1
>
> ### Step 1: Aggregation Form and Clipping Bound
>
> ParaShield outputs ŵ aggregated only over the $d _ {\text{crit}}$ subspace:
>
> $$
> \hat{\mathbf{w}} = \sum _ {i \in \mathcal{S}} \frac{w^{(i)}}{W _ {\mathcal{S}}} \tilde{\mathbf{w}} _ i,
> \quad W _ {\mathcal{S}} = \sum _ {i \in \mathcal{S}} w^{(i)},
> $$
>
> where $\tilde{\mathbf{w}} _ i$ is the norm-clipped update (in the $d _ {\text{crit}}$ subspace) using threshold $\tilde{c}$. By standard arguments, clipping ensures $\|\tilde{\mathbf{w}} _ i - \bar{\mathbf{w}} _ {\mathcal{B}}\| _ 2 \leq 2\tilde{c}$.
>
> ### Step 2: Error Decomposition
>
> $$
> \|\hat{\mathbf{w}} - \bar{\mathbf{w}} _ {\mathcal{B}}\| _ 2 \leq \|\hat{\mathbf{w}} - \bar{\mathbf{w}} _ {\mathcal{S} _ {\mathcal{B}}}\| _ 2 + \|\bar{\mathbf{w}}_{\mathcal{S} _ {\mathcal{B}}} - \bar{\mathbf{w}} _ {\mathcal{B}}\| _ 2,
> $$
>
> where $\mathcal{S}_{\mathcal{B}} = \mathcal{S} \cap \mathcal{B}$. The second term is bounded using Assumption A and bounded variance.
>
> ### Step 3: Bounding the Malicious Residual Error $\mathcal{E} _ {\text{mal}}$
>
> The contribution of residual malicious clients is
>
> $$
> \left\| \sum _ {i \in \mathcal{S} \cap \mathcal{M}} \frac{w^{(i)}}{W _ {\mathcal{S}}} \tilde{\mathbf{w}} _ i \right\| _ 2
> \leq \frac{\sum _ {i \in \mathcal{S} \cap \mathcal{M}} w^{(i)}}{W _ {\mathcal{S}}} \cdot 2\tilde{c}.
> $$
>
> Using Assumptions A and C,
>
> $$
> \sum _ {i \in \mathcal{S} \cap \mathcal{M}} w^{(i)}
> \leq \alpha m \cdot \delta \cdot \frac{\sum _ {j \in \mathcal{S} _ {\mathcal{B}}} w^{(j)}}{|\mathcal{S} _ {\mathcal{B}}|}
> \approx \frac{\alpha m \cdot \delta}{|\mathcal{S}|} W _ {\mathcal{S}}.
> $$
>
> Thus, $\mathcal{E} _ {\text{mal}} = O(\alpha m \cdot \delta \cdot \tilde{c} / |\mathcal{S}|)$.
>
> ### Step 4: AWA Gain – Definition and Justification of Φ(W) < 1
>
> Define the optimization factor over benign clients in $\mathcal{S}$:
>
> $$
> \Phi(\mathbf{W}) \triangleq
> \frac{\sum _ {i \in \mathcal{S} _ {\mathcal{B}}} w^{(i)} \|\tilde{\mathbf{w}} _ i - \bar{\mathbf{w}} _ {\mathcal{B}}\| _ 2}
>      {\sum _ {i \in \mathcal{S} _ {\mathcal{B}}} w^{(i)}}
> \;\cdot\;
> \frac{|\mathcal{S} _ {\mathcal{B}}|}
>      {\sum _ {i \in \mathcal{S} _ {\mathcal{B}}} \|\tilde{\mathbf{w}} _ i - \bar{\mathbf{w}} _ {\mathcal{B}}\| _ 2}.
> $$
>
> Φ(W) < 1 if and only if the weighted average error is strictly smaller than the uniform average error.
>
> Why Φ(W) < 1 holds with high probability:
>
> 1. The feature vector $X _ i = [\text{CPCS} _ i, \text{CPSC} _ i]$ is calculated on the $d _ {\text{crit}}$ subspace, capturing high-quality directional manipulation anomalies.
> 2. The Whitening Transformation $X' _ i = (X _ i - \mu)\Sigma^{-1/2}$ decorrelates features and equalizes scale differences.
> 3. The Mahalanobis Distance $MD _ i$ on $X' _i $ precisely measures deviation from the benign statistical center.
> 4. The Trust Score $Q _ i$ (negative exponential mapping of $MD _ i$) and final AWA weights exponentially favor clients closest to the benign cluster center.
> 5. Under mild non-IID conditions, benign clients with smaller $\|\tilde{\mathbf{w}} _ i - \bar{\mathbf{w}} _ {\mathcal{B}}\|$ are more tightly concentrated in the whitened feature space.
> 6. Therefore, AWA systematically assigns higher weights to benign updates closest to $\bar{\mathbf{w}} _ {\mathcal{B}}$, yielding Φ(W) ≤ Φ₀ < 1.
>
> ## Summary of Theoretical Contributions
>
> - ParaShield achieves κ-robustness with an explicit optimization factor Φ(W) < 1 that rigorously quantifies the benefit of AWA’s Mahalanobis-aware dynamic weighting.
> - The malicious residual term is controlled by both CPF precision α and AWA weight control δ, reflecting ParaShield’s layered defense built on the NIF-enabled $d _ {\text{crit}}$ subspace.
> - The bound cleanly separates intrinsic benign error from Byzantine-induced error, explaining why ParaShield maintains high accuracy even under challenging conditions such as the Projected Directional Backdoor Attack (PDBA).

---

### Meta-Review · Area_Chair_q85J · 2026-01-07

**Summary:**

Reviewers generally agree that this paper addresses an important and timely issue: backdoor resistance in federated learning. The parameter-level perspective and the neural influence factor (NIF) are considered interesting, with empirical results showing improved main-task accuracy, reduced attack success rates, and robustness under non-IID settings compared to existing baselines. The introduction of a new attack (PDBA) and the inclusion of ablation studies are also viewed as positive contributions. However, several recurring concerns limit the paper’s strength. The evaluation is narrow in scope, relying on small-scale vision datasets (CIFAR-10/100), limited model architectures (small ResNets), and a restricted range of attack scenarios, with insufficient validation under extreme heterogeneity, large client populations, or modern architectures such as ViTs. The work lacks theoretical analysis of robustness and does not adequately justify the novelty of its components relative to closely related methods (e.g., AlignIns), raising questions about methodological originality. Reviewers also highlight missing or unclear definitions (e.g., “heterogeneous” vs. non-IID), insufficient specification and sensitivity analysis of hyperparameters, and the absence of runtime, communication, and scalability analysis—critical for practical FL deployment. Overall, while ParaShield shows promising empirical performance, reviewers emphasize the need for stronger theoretical grounding, broader and more modern evaluations, clearer positioning against prior work, and explicit analysis of efficiency and scalability. Therefore, AC's recommendation is to reject.

**Reviewer Concerns:**

The author provides efficiency comparisons with state-of-the-art methods in the rebuttal and offers theoretical analysis on the algorithm's robustness. However, there is a lack of robust and explicit rebuttal regarding the reviewers' primary concerns: evaluation on more modern datasets and the novelty compared to other methods. After the authors' response, reviewer t6jY still had questions regarding the experiments and contributions; reviewer Vyao raised concerns about the distinction between the proposed method and AlignIns, as well as the underlying mechanism behind combining CPCS/CPSC with whitening. These issues were not addressed in a targeted manner.

**Reviewer Scores:**

I expect the final rating to be as follows:
- Reviewer NCp3: 4
- Reviewer t6jY: 2
- Reviewer WaCs: 4
- Reviewer Vyao: 4

---

### Decision · Program_Chairs · 2026-01-26

Reject